# FLAMBE: Structural Complexity and Representation Learning of Low Rank MDPs

**Alekh Agarwal**
Microsoft Research, Redmond
alekha@microsoft.com

**Sham Kakade**
Microsoft Research, NYC
sham@cs.washington.edu

**Akshay Krishnamurthy**
Microsoft Research, NYC
akshaykr@microsoft.com

**Wen Sun**
Cornell University
ws455@cornell.edu

## Abstract

In order to deal with the curse of dimensionality in reinforcement learning (RL), it is common practice to make parametric assumptions where values or policies are functions of some low dimensional feature space. This work focuses on the representation learning question: how can we *learn* such features? Under the assumption that the underlying (unknown) dynamics correspond to a low rank transition matrix, we show how the representation learning question is related to a particular *non-linear* matrix decomposition problem. Structurally, we make precise connections between these low rank MDPs and latent variable models, showing how they significantly generalize prior formulations, such as block MDPs, for representation learning in RL. Algorithmically, we develop FLAMBE, which engages in exploration and representation learning for provably efficient RL in low rank transition models. On a technical level, our analysis eliminates reachability assumptions that appear in prior results on the simpler block MDP model and may be of independent interest.

## 1 Introduction

The ability to learn effective transformations of complex data sources, sometimes called representation learning, is an essential primitive in modern machine learning. Advances in this area have led to remarkable achievements in language modeling, vision, and serve as a partial explanation for the success of deep learning more broadly (Bengio et al., 2013). In Reinforcement Learning (RL), several works have shown empirically that learning succinct representations of perceptual inputs can accelerate the search for decision-making policies (Bellemare et al., 2016; Pathak et al., 2017; Tang et al., 2017; Oord et al., 2018; Srinivas et al., 2020). However, representation learning for RL is far more subtle than it is for supervised learning (Du et al., 2019a; Van Roy and Dong, 2019; Lattimore and Szepesvari, 2020), and the theoretical foundations of representation learning for RL are nascent.

The first question that arises in this context is: what is a good representation? Intuitively, a good representation should help us achieve greater sample efficiency on downstream tasks. For supervised learning, several theoretical works adopt the perspective that a good representation should permit simple models to achieve high accuracy on tasks of interest (Baxter, 2000; Maurer et al., 2016; Arora et al., 2019; Tosh et al., 2020). Lifting this perspective to reinforcement learning, it is natural to expect that we can express value functions and policies as simple functions of our representation. This may allow us to leverage recent work on sample efficient RL with parametric function approximation.

The second question is: how do we *learn* such a representation when it is not provided in advance? This question is particularly challenging because representation learning is intimately tied to explo-

| Algorithm | Setting | Sample Complexity | Computation |
|---|---|---|---|
| PCID (Du et al., 2019b) | block MDP | $d^4 H^2 K^4 \left( \frac{1}{\eta^4 \gamma^2} + \frac{1}{\varepsilon^2} \right)$ | Oracle efficient |
| HOMER (Misra et al., 2020) | block MDP | $d^8 H^4 K^4 \left( \frac{1}{\eta^3} + \frac{1}{\varepsilon^2} \right)$ | Oracle efficient |
| OLIVE (Jiang et al., 2017) | low Bellman rank | $\frac{d^2 H^3 K}{\varepsilon^2}$ | Inefficient |
| Sun et al. (2019) | low Witness rank | $\frac{d^2 H^3 K}{\varepsilon^2}$ | Inefficient |
| FLAMBE (this paper) | low rank MDP | $\frac{d^7 K^9 H^{22}}{\varepsilon^{10}}$ | Oracle efficient |

Table 1: Comparison of methods for representation learning in RL. Settings from least to most general are: block MDP, low rank MDP, low Bellman rank, low Witness rank. In all cases $d$ is the embedding dimension, $H$ is the horizon, $K$ is the number of actions, $\eta$ and $\gamma$ parameterize reachability and margin assumptions, and $\varepsilon$ is the accuracy. Dependence on function classes and logarithmic factors are suppressed. Oracle and realizability assumptions vary. Block MDP algorithms discover a *one-hot* representation to discrete latent states. Bellman/Witness rank approaches can take a class $\Phi$ of embedding functions and search over simple policies or value functions composed with $\Phi$ (see Section 4 and Appendix A.3 for details).

ration. We cannot learn a good representation without a comprehensive dataset of experience from the environment, but a good representation may be critical for efficient exploration.

This work considers these questions in the context of low rank MDPs (Jiang et al., 2017) (also known as factorizing MDPs (Rendle et al., 2010), factored linear MDPs (Yao et al., 2014), and linear MDPs (Jin et al., 2020b; Yang and Wang, 2020)), which we argue provide a natural framework for studying representation learning in RL. Concretely, these models assume there exists low dimensional embedding functions $\phi(x, a), \mu(x')$ such that the transition operator $T$ satisfies
$T(x' \mid x, a) = \langle \phi(x, a), \mu(x') \rangle$, where $T(x' \mid x, a)$ specifies the probability of the next state $x'$ given the previous state $x$ and action $a$. Low rank MDPs address the first issue above (on what constitutes a good representation) in that if the features $\phi$ are known to the learner, then sample efficient learning is possible (Jin et al., 2020b; Yang and Wang, 2020).

**Our contributions.** We address the question of learning the representation $\phi$ in a low rank MDP. To this end our contributions are both structural and algorithmic.

1. **Expressiveness of low rank MDPs.** We first provide a re-formulation of the low rank dynamics in terms of an equally expressive, but more interpretable latent variable model. We provide several structural results for low rank MDPs, relating it to other models studied in prior work on representation learning for RL. In particular, we show that low rank MDPs are significantly more expressive than the block MDP model (Du et al., 2019b; Misra et al., 2020).

2. **Feature learning.** We develop a new algorithm, called FLAMBE for "Feature learning and model based exploration", that learns a representation for low rank MDPs. We prove that under realizability assumptions, FLAMBE learns a *uniformly accurate* model of the environment as well as a feature map that enables the use of linear methods for RL, in a statistically and computationally efficient manner. These guarantees enable downstream reward maximization, for *any* reward function, with no additional data collection.

Our results and techniques provide new insights on representation learning for RL and also significantly increase the scope for provably efficient RL with rich observations (see Table 1).

## 2  Low Rank MDPs

We consider an episodic Markov decision process $\mathcal{M}$ with episode length $H \in \mathbb{N}$, state space $\mathcal{X}$, and a finite action space $\mathcal{A} = \{1, \ldots, K\}$. In each episode, a trajectory $\tau = (x_0, a_0, x_1, a_1, \ldots, x_{H-1}, a_{H-1}, x_H)$ is generated, where (a) $x_0$ is a starting state, and (b) $x_{h+1} \sim T_h(\cdot \mid x_h, a_h)$, and (c) all actions $a_{0:H-1}$ are chosen by the agent. We assume the starting state is fixed and that there is only one available action at time 0.[1] The operators $T_h : \mathcal{X} \times \mathcal{A} \to \Delta(\mathcal{X})$ denote the (non-stationary) transition dynamics for each time step.

As is standard in the literature, a policy $\pi : \mathcal{X} \to \Delta(\mathcal{A})$ is a (randomized) mapping from states to actions. We use the notation $\mathbb{E}\left[\cdot \mid \pi, \mathcal{M}\right]$ to denote expectations over states and actions observed when executing policy $\pi$ in MDP $\mathcal{M}$. We abuse notation slightly and use $[H]$ to denote $\{0, \ldots, H-1\}$.

**Definition 1.** *An operator $T : \mathcal{X} \times \mathcal{A} \to \Delta(\mathcal{X})$ admits a* low rank decomposition *with dimension $d \in \mathbb{N}$ if there exist two embedding functions $\phi^\star : \mathcal{X} \times \mathcal{A} \to \mathbb{R}^d$ and $\mu^\star : \mathcal{X} \to \mathbb{R}^d$ such that*

$$\forall x, x' \in \mathcal{X}, a \in \mathcal{A} : T(x' \mid x, a) = \langle \phi^\star(x, a), \mu^\star(x') \rangle.$$

*For normalization,[2] we assume that $\|\phi^\star(x, a)\|_2 \leq 1$ for all $x, a$ and for any function $g : \mathcal{X} \to [0, 1]$, $\left\| \int \mu^\star(x) g(x) dx \right\|_2 \leq \sqrt{d}$. An MDP $\mathcal{M}$ is a* low rank MDP *if for each $h \in [H]$, $T_h$ admits a low rank decomposition with dimension $d$. We use $\phi_h^\star, \mu_h^\star$ to denote the embeddings for $T_h$.*

Throughout we assume that $\mathcal{M}$ is a low rank MDP with dimension $d$. Note that the normalization condition on $\mu^\star$ ensures that the Bellman backup operator is well-behaved.

**Function approximation for representation learning.**    We consider state spaces $\mathcal{X}$ that are arbitrarily large, so that some form of function approximation is necessary to generalize across states. For representation learning, it is natural to grant the agent access to two function classes $\Phi \subset \{\mathcal{X} \times \mathcal{A} \to \mathbb{R}^d\}$ and $\Upsilon \subset \{\mathcal{X} \to \mathbb{R}^d\}$ of candidate embeddings, which we can use to identify the true embeddings $(\phi^\star, \mu^\star)$. To facilitate this model selection task, we posit a *realizability* assumption.

**Assumption 1** (Realizability). *We assume that for each $h \in [H]$: $\phi_h^\star \in \Phi$ and $\mu_h^\star \in \Upsilon$.*

We desire sample complexity bounds that scale logarithmically with the cardinality of the classes $\Phi$ and $\Upsilon$, which we assume to be finite. Extensions that permit infinite classes with bounded statistical complexity (e.g., VC-classes) are not difficult.

In Appendix A, we show that the low rank assumption alone, without Assumption 1, is not sufficient for obtaining performance guarantees that are independent of the size of the state space. Hence, additional modeling assumptions are required, and we encode these in $\Phi, \Upsilon$.

**Learning goal.**    We focus on the problem of reward-free exploration (Hazan et al., 2019; Jin et al., 2020a), where the agent interacts with the environment with no reward signal. When considering model-based algorithms, a natural reward-free goal is *system identification*: given function classes $\Phi, \Upsilon$, the algorithm should learn a model $\widehat{\mathcal{M}} := (\widehat{\phi}_{0:H-1}, \widehat{\mu}_{0:H-1})$ that uniformly approximates the environment $\mathcal{M}$. We formalize this with the following performance criteria:

$$\forall \pi, h \in [H] : \mathbb{E}\left[ \left\| \left\langle \widehat{\phi}_h(x_h, a_h), \widehat{\mu}_h(\cdot) \right\rangle - T_h(\cdot \mid x_h, a_h) \right\|_{\mathrm{TV}} \mid \pi, \mathcal{M} \right] \leq \varepsilon. \tag{1}$$

Here, we ask that our model accurately approximates the one-step dynamics from the state-action distribution induced by following *any* policy $\pi$ for $h$ steps in the real environment.

System identification also implies a quantitative guarantee on the learned representation $\widehat{\phi}_{0:H-1}$: we can approximate the Bellman backup of any value function on any data-distribution.

**Lemma 1.** *If $\widehat{\mathcal{M}} = (\widehat{\phi}_{0:H-1}, \widehat{\mu}_{0:H-1})$ satisfies (1), then*

$$\forall h \in [H], V : \mathcal{X} \to [0, 1], \exists \theta : \max_\pi \mathbb{E}\left[ \left| \left\langle \theta, \widehat{\phi}_h(x_h, a_h) \right\rangle - \mathbb{E}\left[V(x_{h+1}) \mid x_h, a_h\right] \right| \mid \pi, \mathcal{M} \right] \leq \varepsilon.$$

Thus, linear function approximation using our learned features suffices to fit the $Q$ function associated with any policy and explicitly given reward.[3] The guarantee also enables dynamic programming techniques for policy optimization. In other words, (1) verifies that we have found a good representation, in a quantitative sense, and enables tractable reward maximization for any known reward function.

## 3  Related work

Low rank models are prevalent in dynamics and controls (Thon and Jaeger, 2015; Littman and Sutton, 2002; Singh et al., 2004). The low rank MDP in particular has been studied in several works in

the context of planning (Barreto et al., 2011; Barreto and Fragoso, 2011), estimation (Duan et al., 2020), and in the generative model setting (Yang and Wang, 2019). Regarding nomenclature, to our knowledge the name *low rank MDP* appears first in Jiang et al. (2017), although Rendle et al. (2010) refer to it as *factorizing MDP*, Yao et al. (2014) call it a factored linear MDP, and Barreto et al. (2011) refer to a similar model as *stochastic factorization*. More recently, it has been called the *linear MDP* by Jin et al. (2020b). We use *low rank MDP* because it highlights the key structural property of the dynamics, and because we study the setting where the embeddings are unknown, which necessitates non-linear function approximation.

Turning to reinforcement learning with function approximation and exploration, a large body of effort focuses on (essentially) linear methods (Yang and Wang, 2020; Jin et al., 2020b; Cai et al., 2020; Modi et al., 2020; Du et al., 2019c; Wang et al., 2019; Agarwal et al., 2020). Closest to our work are the results of Jin et al. (2020b) and Yang and Wang (2020), who consider low rank MDPs with known feature maps $\phi_{0:H-1}^\star$ (Yang and Wang (2020) also assumes that $\mu_{0:H-1}^\star$ is known up to a linear map). These results motivate our representation learning formulation, but, on their own, these algorithms cannot leverage the inductive biases provided by neural networks to scale to rich state spaces.

There are methods for more general, non-linear, function approximation, but these works either (a) require strong environment assumptions such as determinism (Wen and Van Roy, 2013; Du et al., 2020), (b) require strong function class assumptions such as bounded Eluder dimension (Russo and Van Roy, 2013; Osband and Van Roy, 2014), (c) have sample complexity scaling linearly with the function class size (Lattimore et al., 2013; Ortner et al., 2014) or (d) are computationally intractable (Jiang et al., 2017; Sun et al., 2019; Dong et al., 2020). Note that Ortner et al. (2014); Jiang et al. (2015) consider a form of representation learning, abstraction selection, but the former scales linearly with the number of candidate abstractions, while the latter does not address exploration.

**Bellman/Witness rank.** We briefly expand on this final category of computationally inefficient methods. For model-free reinforcement learning, Jiang et al. (2017) give an algebraic condition, in terms of a notion called the Bellman rank, on the environment and a given function approximation class, under which sample efficient reinforcement learning is always possible. Sun et al. (2019) extend the definition to model-based approaches, with the notion of Witness rank. As we will see in the next section, the low rank MDP with a function class derived from $\Phi$ (and $\Upsilon$) admits low Bellman (resp., Witness) rank, and so these results imply that our setting is statistically tractable.

**Block MDPs.** Finally, we turn to theoretical works on representation learning for RL. Du et al. (2019b) introduce the *block MDP* model, in which there is a finite latent state space $\mathcal{S}$ that governs the transition dynamics, and each "observation" $x \in \mathcal{X}$ is associated with a latent state $s \in \mathcal{S}$, so the state is *decodable*. The natural representation learning goal is to recover the latent states, and Du et al. (2019b); Misra et al. (2020) show that this can be done, in concert with exploration, in a statistically and computationally efficient manner. Since the block MDP can be easily expressed as a low rank MDP, our results can be specialized to this setting, where they yield comparable guarantees. On the other hand, we will see that the low rank MDP is significantly more expressive, and so our results greatly expand the scope for provably efficient representation learning and reinforcement learning.

## 4 Expressiveness of low rank MDPs

Before turning to our algorithmic development, we discuss connections between low-rank MDPs and models studied in prior work. This discussion is facilitated by formalizing a connection between MDP transition operators and latent variable graphical models.

**Definition 2.** *The* latent variable representation *of a transition operator* $T : \mathcal{X} \times \mathcal{A} \to \Delta(\mathcal{X})$ *is a latent space* $\mathcal{Z}$ *along with functions* $\psi : \mathcal{X} \times \mathcal{A} \to \Delta(\mathcal{Z})$ *and* $\nu : \mathcal{Z} \to \Delta(\mathcal{X})$*, such that* $T(\cdot \mid x, a) = \int \nu(\cdot \mid z)\psi(z \mid x, a)dz$. *The* latent variable dimension *of* $T$*, denoted* $d_{\mathrm{LV}}$ *is the cardinality of smallest latent space* $\mathcal{Z}$ *for which* $T$ *admits a latent variable representation.*

See Figure 1. In this representation, (1) each $(x, a)$ pair induces a "posterior" distribution $\psi(x, a) \in \Delta(\mathcal{Z})$ over $z$, (2) we sample $z \sim \psi(x, a)$, and (3) then sample $x' \sim \nu(\cdot \mid z)$, where $\nu$ specifies the "emission" distributions. As notation, we typically write $\nu(x) \in \mathbb{R}^{\mathcal{Z}}$ with coordinates $\nu(x)[z] = \nu(x \mid z)$ and we call $\psi, \nu$ the *simplex features*, following the example described by Jin et al. (2020b). When considering $H$-step MDPs, this representation allows us to augment the trajectory $\tau$ with the

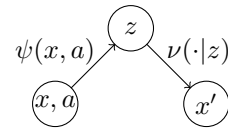

Figure 1: The latent variable interpretation.

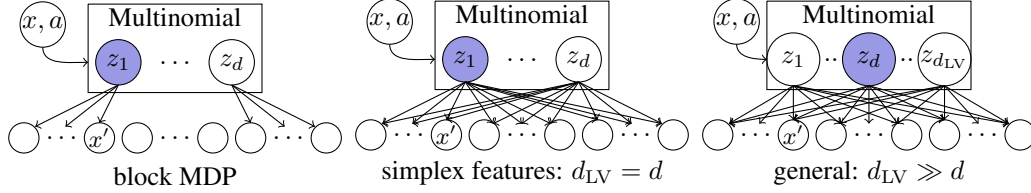

Figure 2: The latent variable interpretation of low rank MDPs, where $(x, a)$ induces a distribution over latent variable $z$. Left: in block MDPs, latent variables induce a partition over the next state $x'$. Center: simplex features have embedding dimension equal to the number of latent variables. Right: low rank MDPs can have exponentially more latent variables than the dimension, $d_{\text{LV}} \gg d$.

latent variables $\tau = (x_0, a_0, z_1, x_1, \ldots z_{H-1}, x_{H-1}, a_{H-1}, x_H)$. Here note that $z_h$ is the latent variable that generates $x_h$.

Note that all transition operators admit a trivial latent variable representation, as we may always take $\psi(x, a) = T(\cdot \mid x, a)$. However, when $T$ is endowed with additional structure, the latent variable representations are more interesting. For example, this viewpoint already certifies a factorization $T(x' \mid x, a) = \langle \psi(x, a), \nu(x') \rangle$ with embedding dimension $|\mathcal{Z}|$, and so $d_{\text{LV}}$ (if it is finite) is an upper bound on the rank of the transition operator. On the other hand, compared with Definition 1, this factorization additionally requires that $\psi(x, a)$ and $\nu(\cdot \mid z)$ are probability distributions. Since the factorization is non-negative, $d_{\text{LV}}$ is the *non-negative rank* of the transition operator.

The latent variable representation enables a natural comparison of the expressiveness of various models, and, as we will see in the next section, yields insights that facilitate algorithm design. We now examine models that have been introduced in prior works and their properties relative to Definition 1.

**Block MDPs.** A block MDP (Du et al., 2019b; Misra et al., 2020) is clearly a latent variable model with $\mathcal{Z}$ corresponding to the latent state space $\mathcal{S}$ and the additional restriction that two latent variables $z$ and $z'$ have disjoint supports in their respective emissions $\nu(\cdot \mid z)$ and $\nu(\cdot \mid z')$ (see the left panel of Figure 2). Therefore, a block MDP is a low rank MDP with rank $d \leq |\mathcal{S}|$, but the next result shows that a low rank MDP is significantly more expressive.

**Proposition 1.** *For any $d \geq 2$ and any $M \in \mathbb{N}$ there exists an environment on $|\mathcal{X}| = M$ states, that can be expressed as a low rank MDP with embedding dimension $d$, but for which any block MDP representation must have $M$ latent states.*

In fact, the MDP that we construct for the proof, admits a latent variable representation with $|\mathcal{Z}| = d$, but does not admit a non-trivial block MDP representation. This separation exploits the decodability restriction of block MDPs, which is indeed quite limiting in terms of expressiveness.

**Simplex features.** Given the latent variable representation and the fact that it certifies a rank of at most $d_{\text{LV}}$, it is natural to ask if this representation is canonical for all low rank MDPs. In other words, for any transition operator with rank $d$, can we express it as a latent variable model with $|\mathcal{Z}| = d$, or equivalently with simplex features of dimension $d$?

As discussed above, this model is indeed more expressive than the block MDP. However, the next result answers the above question in the negative. The latent variable representation is *exponentially* weaker than the general low rank representation in the following sense:

**Proposition 2.** *For any even $n \in \mathbb{N}$, there exists an MDP that can be cast as a low rank MDP with embedding dimension $O(n^2)$, but which has $d_{\text{LV}} \geq 2^{\Omega(n)}$.*

See the center and right panels of Figure 2. The result is proved by recalling that the latent variable dimension determines the non-negative rank of $T$, which can be much larger than its rank (Rothvoß, 2017; Yannakakis, 1991). It showcases how low rank MDPs are quite different from latent variable models of comparable dimension and demonstrates how embedding functions with negative values can provide significant expressiveness.

**Bellman and Witness rank.** As our last concrete connection, we remark here that the low rank MDP with a function class derived from $\Phi$ (and $\Upsilon$) admits low Bellman (resp., Witness) rank.

**Proposition 3** (Informal). *The low rank MDP model always has Bellman rank at most $d$. Additionally, given $\Phi$ and assuming $\phi^\star_{0:H-1} \in \Phi$, we can construct a function classes $(\mathcal{G}, \Pi)$, so that OLIVE when run with $(\mathcal{G}, \Pi)$ has sample complexity $\tilde{O}\left(\text{poly}(d, H, K, \log |\Phi|, \epsilon^{-1})\right)$.*

See Proposition 5 for a more precise statement. An analogous result holds for the Witness rank notion of Sun et al. (2019) (see Proposition 6 in the appendix). Unfortunately both OLIVE, and the algorithm of Sun et al. (2019) are not computationally tractable, as they involve enumeration of the employed function class. We turn to the development of computationally tractable algorithms in the next section.

# 5 Main results

We now turn to the design of algorithms for representation learning and exploration in low rank MDPs. As a computational abstraction, we consider the following optimization and sampling oracles.

**Definition 3** (Computational oracles). *Define the following oracles for the classes $\Phi, \Upsilon$:*

1. *The* maximum likelihood oracle, MLE, *takes a dataset $D$ of $(x, a, x')$ triples, and returns*

$$\text{MLE}(D) := \text{argmax}_{\phi \in \Phi, \mu \in \Upsilon} \sum_{(x,a,x') \in D} \log(\langle \phi(x,a), \mu(x') \rangle).$$

2. *The* sampling oracle, SAMP, *is a subroutine which, for any $(\phi, \mu) \in \Phi \times \Upsilon$ and any $(x, a)$, returns a sample $x' \sim \langle \phi(x,a), \mu(\cdot) \rangle$. Multiple calls to the procedure result in independent samples.*

We assume access to both oracles as a means towards practical algorithms that avoid explicitly enumerating over all functions in $\Phi$ and $\Upsilon$. Related assumptions are quite common in the literature (Misra et al., 2020; Du et al., 2019b; Agarwal et al., 2014), and in practice, both oracles can be reasonably approximated whenever optimizing over $\Phi, \Upsilon$ is feasible (e.g., neural networks). Regarding MLE, other optimization oracles are possible, and in the appendix (Remark 19) we sketch how our proof can accommodate a generative adversarial oracle as a replacement (Goodfellow et al., 2014; Arora et al., 2017). While the sampling oracle is less standard, one might implement SAMP via optimization methods like the Langevin dynamics (Welling and Teh, 2011) or through reparametrization techniques such as the Gumbel-softmax trick (Jang et al., 2017; Figurnov et al., 2018).[4]

## 5.1 Algorithm description

The algorithm is called FLAMBE, for "Feature Learning And Model-Based Exploration." Pseudocode is displayed in Algorithm 1. Before turning to the description, we clarify our use of $h$-step policies and policy mixtures. Throughout the algorithm we compute policies by optimizing reward functions that are only defined at a single stage $\tilde{h} \in [H]$. This yields an $\tilde{h}$-*step policy*, and when executing such policy, we always terminate the episode by taking actions $\{a_h\}_{h>\tilde{h}}$ uniformly at random. We also use *policy mixtures* or distributions over policies. When executing a policy mixture, we sample a policy from the distribution at the beginning of the episode and use that policy for the entire episode.

FLAMBE is iterative in nature, where in iteration $j$, we use an exploration policy $\rho_{j-1}$ to collect a dataset of transitions (Line 6), and then we pass all the transitions collected so far to the MLE oracle (Line 7). The MLE oracle returns embedding functions $(\widehat{\phi}_h, \widehat{\mu}_h)$ for each $h$, which define transition operators $\widehat{T}_h$ for the learned model $\widehat{\mathcal{M}}$. Then FLAMBE calls a planning sub-routine to compute the exploration policy $\rho_j$ (Lines 10- 11) that we use in the next iteration. After $J_{\max}$ iterations, we output the current model $\widehat{\mathcal{M}}$.

For the planning step, intuitively we seek an exploratory policy $\rho$ that induces good coverage over the state space when executed in the model. We do this by solving one planning problem per time step $h$ in Algorithm 2 using a technique inspired by elliptical potential arguments from linear bandits (Dani et al., 2008). Using the $\tilde{h}$-step model $\widehat{T}_{0:\tilde{h}-1}$, we iteratively maximize certain quadratic forms of our learned features $\widehat{\phi}_{\tilde{h}-1}$ to find new directions not covered by the previously discovered policies, and we update the exploratory policy mixture to include the maximizer. The planning algorithm terminates when no policy can achieve large quadratic form, which implies that we have found all reachable directions in $\widehat{\phi}_{h-1}$. This yields a policy mixture $\rho_h^{\text{pre}}$ with component policies that are

**Algorithm 1** FLAMBE: Feature Learning And Model-Based Exploration

---

1: **Input:** Environment $\mathcal{M}$, function classes $\Phi, \Upsilon$, subroutines MLE and SAMP, parameters $\beta, n$.
2: For each $h$ set $\rho_h^{\text{pre}}$ to be the random policy, which takes all actions uniformly at random.
3: Set $\rho_0 = \{\rho_h^{\text{pre}}\}_{h=0}^{H-1}$ and $D_h = \emptyset$ for each $h \in \{0, \ldots, H-1\}$.
4: **for** $j = 1, \ldots, J_{\max}$ **do**
5:     **for** $h = 0, \ldots, H-1$ **do**
6:         Obtain $D_h \leftarrow D_h \cup$ EXECUTE_IN_REAL_WORLD$(\rho_{j-1}, h, n)$.
7:         Solve maximum likelihood problem: $(\widehat{\phi}_h, \widehat{\mu}_h) \leftarrow$ MLE$(D_h)$.
8:         Set $\widehat{T}_h(x_{h+1} \mid x_h, a_h) = \left\langle \widehat{\phi}_h(x_h, a_h), \widehat{\mu}_h(x_{h+1}) \right\rangle$.
9:     **end for**
10:     For each $h$, call planner (Algorithm 2) with $h$ step model $\widehat{T}_{0:h-1}$ and $\beta$ to obtain $\rho_h^{\text{pre}}$.
11:     Set $\rho_j = \{\rho_h^{\text{pre}}\}_{h=0}^{H-1}$ to be the set of exploration policies.
12: **end for**
13: **function** EXECUTE_IN_REAL_WORLD$(\rho = \{\rho_t^{\text{pre}}\}_{t=0}^{H-1}, h, n)$
14:     Initialize $D = \emptyset$.
15:     **for** $i = 1, \ldots, n$ **do**
16:         Pick $t \in \{0, \ldots, H-1\}$ uniformly at random.
17:         Starting in $x_0$, execute $h-1$ actions using $\rho_t^{\text{pre}}$ to get $x_h$. Take $a_h$ uniformly to get $x_{h+1}$.
18:         Augment $D \leftarrow D \cup \{(x_h, a_h, x_{h+1})\}$.
19:     **end for**
20:     **return** $D$.
21: **end function**

---

---

**Algorithm 2** Elliptical planner

---

1: **Input:** MDP $\widetilde{\mathcal{M}} = (\phi_{0:\tilde{h}}, \mu_{0:\tilde{h}})$, subroutine SAMP, parameter $\beta > 0$. Initialize $\Sigma_0 = I_{d \times d}$.
2: **for** $t = 1, 2, \ldots,$ **do**
3:     Use SAMP to compute (see text for details)

$$\pi_t = \arg\max_{\pi} \mathbb{E}\left[ \phi_{\tilde{h}}(x_{\tilde{h}}, a_{\tilde{h}})^\top \Sigma_{t-1}^{-1} \phi_{\tilde{h}}(x_{\tilde{h}}, a_{\tilde{h}}) \mid \pi, \widetilde{\mathcal{M}} \right]. \tag{2}$$

4:     If the objective is at most $\beta$, halt and output $\rho = \text{unif}(\{\pi_\tau\}_{\tau < t})$.
5:     Compute $\Sigma_{\pi_t} = \mathbb{E}\left[ \phi_{\tilde{h}}(x_{\tilde{h}}, a_{\tilde{h}}) \phi_{\tilde{h}}(x_{\tilde{h}}, a_{\tilde{h}})^\top \mid \pi, \widetilde{\mathcal{M}} \right]$. Update $\Sigma_t \leftarrow \Sigma_{t-1} + \Sigma_{\pi_t}$.
6: **end for**

---

linear in the learned features $\widehat{\phi}_{0:h-1}$. The challenge in our analysis is to relate this coverage in the model to that in the true environment as we discuss in the next section.

Algorithm 2 is a model-based planner, so it requires no interaction with the environment. The main computational step is the optimization problem (2), which is a policy optimization problem in a known low rank MDP from which we can sample efficiently. We solve this problem by running the reinforcement learning algorithm of Jin et al. (2020b) (See Lemma 14 in the appendix) with the reward function corresponding to the objective in (2) and using SAMP to simulate the algorithm's interaction with the environment. Note that we are optimizing over *all policies*, which is possible because the Bellman backups in a low rank MDP are linear functions of the features (c.f., Lemma 1). The sampling oracle can also be used to approximate all expectations, and, with sufficient accuracy, this has no bearing on the final results. Our proofs do account for the sampling errors.

## 5.2 Theoretical Results

We now state the main guarantee.

**Theorem 2.** *Fix $\delta \in (0, 1)$. If $\mathcal{M}$ is a low rank MDP with dimension $d$ and horizon $H$ and Assumption 1 holds, then* FLAMBE *with subroutine Algorithm 2 and appropriate settings[5] of $\beta$, $J_{\max}$, and $n$, computes a model $\widehat{\mathcal{M}}$ such that (1) holds with probability at least $1 - \delta$. The total number of*

*trajectories collected is*

$$\tilde{O}\left(\frac{H^{22}K^9 d^7 \log(|\Phi||\Upsilon|/\delta)}{\varepsilon^{10}}\right).$$

*The algorithm runs in polynomial time with polynomially many calls to* MLE *and* SAMP *(Definition 3).*

Thus, FLAMBE provably learns low rank MDP models in a statistically and computationally efficient manner, under Assumption 1. While the result is comparable to prior work in the dependencies on $d$, $H$, $K$ and $\varepsilon$, we instead highlight the more conceptual advances over prior work.

- The key advancement over the block MDP algorithms (Du et al., 2019b; Misra et al., 2020) is that FLAMBE applies to a significantly richer class of models with comparable function approximation assumptions.[6] A secondary, but important, improvement is that FLAMBE *does not require any reachability assumptions*, unlike these previous results. We remark that Feng et al. (2020) avoid reachability restrictions in block MDPs, but their function approximation/oracle assumptions are much stronger than ours.[7]

- Over Jin et al. (2020b); Yang and Wang (2020), the key advancement is that we address the representation learning setting where the embeddings $\phi^\star_{0:H-1}$ are not known a priori. On the other hand, our bound scales with the number of actions $K$. We believe that additional structural assumptions on $\Phi$ are required to avoid the dependence on $K$ in the representation learning setting.

- Over Jiang et al. (2017); Sun et al. (2019), the key advancement is computational efficiency. However, the low rank MDP is less general than what is covered by their theory, and our sample complexity is worse in the polynomial factors.

As remarked earlier, the logarithmic dependence on the sizes of $\Phi, \Upsilon$ can be relaxed to alternative notions of capacity for continuous classes.

We also state a sharper bound for a version of FLAMBE that operates directly on the simplex factorization. The main difference is that we use a conceptually simpler planner (See Algorithm 3 in the appendix) and the sample complexity bound scales with $d_{\mathrm{LV}}$.

**Theorem 3.** *Fix $\delta \in (0,1)$. If $\mathcal{M}$ admits a simplex factorization with embedding dimension $d_{\mathrm{LV}}$, Assumption 1 holds, and all $\phi \in \Phi$ satisfy $\phi(x,a) \in \Delta([d_{\mathrm{LV}}])$, then* FLAMBE *with Algorithm 3 as the subroutine and appropriate setting[8] of $J_{\max}$ and $n$ computes a model $\widehat{\mathcal{M}}$ such that (1) holds with probability at least $1 - \delta$. The total number of trajectories collected is*

$$\tilde{O}\left(\frac{H^{11}K^5 d_{\mathrm{LV}}^5 \log(|\Phi||\Upsilon|/\delta)}{\varepsilon^3}\right).$$

*The algorithm runs in polynomial time with polynomially many calls to* MLE *and* SAMP *(Definition 3).*

This bound scales much more favorably with $H$, $K$ and $\varepsilon$, but incurs a polynomial dependence on the latent variable dimension $d_{\mathrm{LV}}$, instead of the embedding dimension $d$. For many problems, including block MDPs, we expect that $d_{\mathrm{LV}} \approx d$, in which case using this version of FLAMBE may be preferable. However, Theorem 3 requires that we encode simplex constraints into our function class $\Phi$, for example using the softmax. When $d_{\mathrm{LV}}$ is small, this may be a practically useful design choice.

**Challenges in the analysis.** We highlight three main challenges in the analysis. The first challenge arises in the analysis of the model learning step, where we want to show that our model $\widehat{T}_h$, learned by maximum likelihood estimation, accurately approximates the true dynamics $T_h$ on the data distributions induced by the exploratory policies $\rho_{0:j-1}$. This requires a generalization argument, but both the empirical MLE objective and population version — the KL divergence — are unbounded, so we cannot use standard uniform convergence techniques. Instead, we employ (and slightly adapt)

results from the statistics literature ([Van de Geer](#), 2000; [Zhang](#), 2006) to show that MLE yields convergence in the Hellinger and total variation distances. While these arguments are well-known in the statistics community, we highlight them here because we believe they may be broadly useful in the context of model-based RL.

The second challenge is to transfer the model error guarantee from the exploratory distributions to distributions induced by other policies. Intuitively, error transfer should be possible if the exploratory policies cover the state space, but we must determine how we measure and track coverage. Leveraging the low rank dynamics, we show that if a policy $\pi$ induces a distribution over true features $\phi_{h-1}^\star$ that is in the span of the directions visited by the exploratory policies, then we can transfer the MLE error guarantee for $\widehat{T}_h$ to $\pi$'s distribution. This suggests measuring coverage for time $h$ in terms of the second moment matrix of the true features at the previous time induced by the exploratory policies $\rho$, that is $\Sigma_{h-1} = \mathbb{E}_\rho \phi_{h-1}^\star \phi_{h-1}^{\star,\top}$. Using these matrices, we prove a sharp *simulation lemma* that bounds the model error observed by a policy $\pi$ in terms of the model error on the exploratory distribution and the probability that $\pi$ visits features for which $\phi_{h-1}^{\star,\top} \Sigma_{h-1}^{-1} \phi_{h-1}^\star$ is large.

The simulation lemma suggests a planning strategy and an "explore-or-terminate" argument: the next exploratory policy should maximize $\phi_{h-1}^{\star,\top} \Sigma_{h-1}^{-1} \phi_{h-1}^\star$, in which case either we visit some new direction and make progress, or we certify [(1)](#), since no policy can make this quantity large. However, we cannot maximize this objective directly, since we do not know $\phi_{h-1}^\star$ or $\Sigma_{h-1}$! Moreover, we cannot use $\widehat{\phi}_{h-1}$ to approximate $\phi_{h-1}^\star$ in the objective, since even if the model is accurate, the features may not be. Instead we plan to find a policy that induces a well-conditioned covariance matrix in terms of $\widehat{\phi}_{h-2}$, the learned features at time $h-2$. By composing this policy with a random action and applying our simulation lemma, we can show that this policy either explores at some previous time or approximately maximizes $\phi_{h-1}^{\star,\top} \Sigma_{h-1}^{-1} \phi_{h-1}^\star$, which allows us to apply the explore-or-terminate argument. Note that this argument uses two random actions: we take $a_{h-1}$ at random to approximately maximize $\phi_{h-1}^{\star,\top} \Sigma_{h-1}^{-1} \phi_{h-1}^\star$, even though we optimize with $\widehat{\phi}_{h-2}$, and then we take $a_h$ at random for the MLE step. Combining this reasoning with an elliptical potential argument, we can bound the number of iterations in which exploration can happen, which leads to the final result.

## 6  Discussion

This paper studies representation learning and exploration for low rank MDPs. We provide an intuitive interpretation of these models in terms of a latent variable representation, and we prove a number of structural results certifying that low rank MDPs are significantly more expressive than models studied in prior work. We also develop FLAMBE, a computationally and statistically efficient model-based algorithm for system identification in low rank MDPs. Policy optimization follows as a corollary.

Our results raise a number of promising directions for future work. On the theoretical side, in the reward-sensitive setting, can we avoid learning the entire model, perhaps by utilizing "value-aware" methods ([Farahmand et al.](#), 2017; [Ayoub et al.](#), 2020)? Can we remove realizability conditions on $\mu_{0:H-1}^\star$ and develop provably efficient model-free algorithms for representation learning in the low rank MDP? On the empirical side, can we leverage the algorithmic insights of FLAMBE to develop new practically effective representation learning algorithms for complex reinforcement learning tasks? Finally, this work considers representation learning for a set of related RL problems which all share the same underlying dynamics, but differ in their reward functions. A natural future direction is to consider learning representations that enable more general task variations. We look forward to answering these questions in future work.

## Broader impact

This paper is theoretical in nature, and so we expect the ethical and societal consequences of our specific results to be minimal. More broadly, we do expect that reinforcement learning will have significant impact on society. There is much potential for benefits to humanity in the often-referenced application domains of precision medicine, personalized education, and elsewhere. There is also much potential for harms, both malicious and unintentional. To this end, we hope that research into the foundations of reinforcement learning can help enable these applications and mitigate harms through the development of algorithms that are efficient, robust, and safe.

## Acknowledgments and Disclosure of Funding

We thank Ruosong Wang for insightful discussions about removing reachability restrictions. Funding in direct support of this work was provided by Microsoft. SK gratefully acknowledges funding from the ONR award N00014-18-1-2247 and NSF Awards CCF-1703574 and CCF-1740551. AK gratefully acknowledges funding from NSF Award IIS-1763618. This work was completed while WS was at Microsoft Research. Additional relevant institutions: University of Washington, University of Massachusetts Amherst, Cornell University.

## Footnotes

[1]This easily accommodates the standard formulation with a non-degenerate initial distribution by defining $T_0(\cdot \mid x_0, a_0)$ to be the initial distribution. This setup is notationally more convenient, since we do not need special notation for the starting distribution.

[2]See the proof of Lemma B.1 in Jin et al. (2020b) for this form of the normalization assumption.

[3]Formally, we append the immediate reward to the learned features $\widehat{\phi}_{0:H-1}$.

[4]We do not explicitly consider approximate oracles, but additive approximations can be accommodated in our proof. In particular, if SAMP returns a sample from a distribution that is $\varepsilon_{\text{samp}}$ close in total variation to the target distribution in $\text{poly}(1/\varepsilon_{\text{samp}})$ time, then we retain computational efficiency.

[5]The precise settings for $\beta$, $J_{\max}$, and $n$ are given in the appendix.

[6]The realizability assumption for Block MDPs implies realizability of $\phi^\star_{0:H-1}$ and the support of the components of $\mu^\star_{0:H-1}$ (recall that for Block MDPs these components have disjoint support). This is slightly weaker than Assumption 1, but stronger than just assuming realizability of $\phi^\star_{0:H-1}$.

[7]Unlike ours, their oracle assumption is not a purely computational abstraction, but rather it implicitly places statistical restrictions on the emission distributions.

[8]This version does not require the parameter $\beta$.

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
