[Supplementary Material]

# A Proofs for the structural results

In this appendix we provide proofs for the structural results in the paper. We first provide the proof of Lemma 1. In Appendix A.2 we focus on results relating to realizability. Then in Appendix A.3 we turn to the separation results of Proposition 1 and Proposition 2. Finally in Appendix A.4 we provide details about the connection to the Bellman and Witness rank.

## A.1 Proof of Lemma 1

*Proof of Lemma 1.* Fix $h$ and $V : \mathcal{X} \to [0,1]$. We drop the dependence on $h$ from the notation, with $x, a$ always corresponding to states and actions at time $h$ and $x'$ corresponding to an action at time $h+1$. Observe that as $\widehat{\mathcal{M}}$ is a low rank MDP, we have

$$\forall x, a : \quad \mathbb{E}\left[V(x') \mid x, a, \widehat{\mathcal{M}}\right] = \left\langle \widehat{\phi}(x,a), \int \widehat{\mu}(x') V(x') \right\rangle =: \left\langle \widehat{\phi}(x,a), \theta \right\rangle$$

Combining with (1), we have that for any policy $\pi$:

$$\mathbb{E}\left[\left|\left\langle \widehat{\phi}(x,a), \theta \right\rangle - \mathbb{E}\left[V(x') \mid x, a, \mathcal{M}\right]\right| \mid \pi, \mathcal{M}\right]$$
$$= \mathbb{E}\left[\left|\mathbb{E}\left[V(x') \mid x, a, \widehat{\mathcal{M}}\right] - \mathbb{E}\left[V(x') \mid x, a, \mathcal{M}\right]\right| \mid \pi, \mathcal{M}\right]$$
$$\leq \mathbb{E}\left[\left\|\left\langle \widehat{\phi}(x,a), \widehat{\mu}(\cdot) \right\rangle - T(\cdot \mid x, a)\right\|_{\mathrm{TV}} \mid \pi, \mathcal{M}\right] \leq \varepsilon. \qquad \square$$

## A.2 On realizability

**Proposition 4.** *Fix $M \in \mathbb{N}$, $n \leq {}^M/_2$, and any algorithm. There exists a low rank MDP over $M$ states with rank $2$ and horizon $2$ such that, if the algorithm collects $n$ trajectories and outputs a policy $\widehat{\pi}$, then with probability at least ${}^1/_8$, $\widehat{\pi}$ is at least ${}^1/_8$-suboptimal for the MDP.*

The result shows that if the low rank MDP has $M$ states, then we require $n = \Omega(M)$ samples to find a near-optimal policy with moderate probability. Thus low rank structure alone is not sufficient to obtain sample complexity guarantees that are independent of the number of states.

In fact the proof yields a slightly stronger conclusion, as the embedding $\mu^\star$ is actually known to the algorithm in the construction. Thus, even when $\mu^\star$ is known, we require some inductive bias on $\phi^\star$ to obtain sample complexity scaling independently with the number of states.

*Proof of Proposition 4.* The result is obtained by embedding a binary classification problem into a low rank MDP and appealing to a standard binary classification lower bound argument. We construct a family of one-step transition operators, all of which have rank $2$. The state space at the current time is of size $M$ and there are two actions $\mathcal{A} := \{0, 1\}$. From each $(x, a)$ pair we transition deterministically either to $x_g$ or $x_b$, and we receive reward $1$ from $x_g$ and reward $0$ from $x_b$.

Formally, we denote the states as $\{x_1, \ldots, x_M\}$ and index each instance by a binary vector $v \in \{0, 1\}^M$, which specifies the good action for each state. The transition operator is

$$T_v(\cdot \mid x_j, a) = \begin{cases} x_g & \text{if } a = v_j \\ x_b & \text{if } a \neq v_j \end{cases}$$

There are therefore $2^M$ instances. Note that as there are only two states at the next time, we trivially see that that transition operator of each instance is rank $2$ and the low rank MDP representation is:

$$\phi_v^\star(x_j, a) = (\mathbf{1}\{a = v_j\}, \mathbf{1}\{a \neq v_j\}) \quad \text{and} \quad \mu_v^\star(x') = (\mathbf{1}\{x' = x_g\}, \mathbf{1}\{x' = x_b\}).$$

Note that $\mu^\star = \mu_v^\star$ does not actually depend on the instance index $v$, so $\mu^\star$ is known to the algorithm for the purposes of the lower bound.

The starting distribution is uniform over $[M]$, so that in $n$ episodes, the agent collects a dataset $\{(x^{(i)}, a^{(i)}, y^{(i)})\}_{i=1}^n$ where $x^{(i)} \sim \mathrm{unif}(x_1, \ldots, x_M)$, $a^{(i)}$ is chosen by the agent and $y^{(i)}$ denotes whether the agent transitions to $x_g$ or $x_b$. Information theoretically, this is equivalent to obtaining $n$ samples from the following data generating process: sample $j \in \mathrm{unif}([M])$ and reveal $v_j$.

In this latter process, we can apply a standard binary classification lower bound argument. Let $P_v$ denote the data distribution where indices $j$ are sampled uniformly at random and labeled by $v_j$. Let $P_v^{(n)}$ denote the product measure where $n$ samples are generated iid from $P_v$. By randomizing the instance, for any example that does not appear in the sample, the probability of error is $1/2$. Therefore the probability of error for any classifier $\widehat{f}$ is

$$
\max_v \mathbb{E}_{S \sim P_v^{(n)}} \mathbb{P}_{j \sim \mathrm{unif}([M])}[\widehat{f}(j) \neq v_j] \geq \mathbb{E}_{v \sim \mathrm{unif}(\{0,1\}^M)} \mathbb{E}_{S \sim P_v^{(n)}} \mathbb{P}_{j \sim \mathrm{unif}([M])}[\widehat{f}(j) \neq v_j]
$$

$$
= \frac{1}{M} \sum_{j=1}^{M} \mathbb{E}_v \mathbb{E}_{S \sim P_v^{(n)}} \mathbf{1}\{\widehat{f}(j) \neq v_j\} \geq \frac{1}{M} \sum_{j=1}^{M} \frac{1}{2} \mathbb{P}_S[j \notin S]
$$

$$
= \frac{1}{2} \left(1 - \frac{1}{M}\right)^n \geq \frac{1}{2}\left(1 - n/M\right).
$$

The second inequality uses the fact that if $j$ does not appear in the sample then $v_j \sim \mathrm{Ber}(1/2)$. Equivalently, we can first sample $n$ unlabeled indices, then commit to the label just on these indices, so that the label for any index not in the sample remains random. Thus for any classifier, there exists some instance for which on average over the sample, the probability of error is at least $1/4$ as long as $n \leq M/2$. This also implies that with constant probability over the sample the error rate is at least $1/8$, since for any random variable $Z$ taking values in $[0, 1]$, we have

$$
\mathbb{E}[Z] \leq 1/8\left(1 - \mathbb{P}[Z > 1/8]\right) + \mathbb{P}[Z \geq 1/8] \leq 1/8 + \mathbb{P}[Z \geq 1/8].
$$

Taking $Z = \mathbb{P}_j[\widehat{f}(j) \neq v_j]$, we have

$$
\mathbb{P}_{S \sim P_v^{(n)}} \left[\mathbb{P}_j[\widehat{f}_j \neq v_j] \geq 1/8\right] \geq \frac{1}{2}\left(1 - n/M\right) - 1/8 \geq 1/8,
$$

where the last inequality holds with $n \leq M/2$.

Now, notice that we can identify any predictor with a policy in the obvious way and also that the sub-optimality for a policy is precisely the classification error for the predictor. With this correspondence, we obtain the result. $\square$

## A.3 Separation results

*Proof of Proposition 1.* Fix $N$ and consider an MDP with horizon 2, where at stage 1 there is only one state $x$ and two actions $a_1, a_2$. At stage 2 there are $N$ possible states, so that $T(\cdot \mid x, a_i) \in \Delta([N])$ for each $i \in \{1, 2\}$. We define the transition operator for stage 1, called $T$ for brevity, explicitly in terms of its factorization. Let $\phi(x, a_1) = e_1$ and $\phi(x, a_2) = e_2$ where $e_1, e_2 \in \mathbb{R}^2$ denotes the two standard basis elements in two dimensions. We define $\mu_1(i) = 1/N$, $\mu_2(i) = i/(\sum_{j=1}^{N} j)$ and $\mu(i) = (\mu_1(i), \mu_2(i)) \in \mathbb{R}^2$. Thus $T(x' = i \mid x, a) = \langle \phi(x, a), \mu(i) \rangle$, which can be easily verified to be a valid transition operator. By construction $T$ has rank 2.

For clarity we express $T$ as the $2 \times N$ matrix.

$$
T := \begin{pmatrix} 1/N & 1/N & \cdots & 1/N \\ 1/(\sum_{j=1}^{N} j) & 2/(\sum_{j=1}^{N} j) & \cdots & N/(\sum_{j=1}^{N} j) \end{pmatrix}.
$$

We now show that the block MDP representation must have $N$ latent states. Suppose the block MDP representation is $T(x' = i \mid x, a) = \langle \phi_B(x, a), \mu_B(i) \rangle$. The block MDP representation requires that for each index $i$ the vector $\mu_B(i)$ is one-sparse. From this, we deduce a constraint that arises when two states belong to the same block. If $i, j$ belong to the same block, say block $b$, then for each $(x, a) \in \mathcal{X} \times \mathcal{A}$, we have

$$
T(x' = i \mid x, a) = \phi_B(x, a)[b]\mu_B(i)[b] = \frac{\mu_B(i)[b]}{\mu_B(j)[b]} \cdot \phi_B(x, a)[b]\mu_B(j)[b]
$$

$$
= \frac{\mu_B(i)[b]}{\mu_B(j)[b]} \cdot T(x' = j \mid x, a)
$$

In words, if states $i, j$ at stage 2 belong to the same block, then the vectors $T(x' = i \mid \cdot), T(x' = j \mid \cdot)$ must be pairwise linearly dependent.[9] Based on our construction, $T(x' = i \mid \cdot) = \mu(i)$, which is just the $i^{\text{th}}$ column of the matrix $T$. By inspection, all $N$ vectors are pairwise linearly independent, and so we can conclude that the block MDP representation must have $N$ latent states. $\quad\square$

*Proof of Proposition 2.* We consider a one step transition operator $T$ that we instantiate to be the slack matrix describing a certain polyhedral set. Let $n$ be even and let $K_n$ be the complete graph on $n$ vertices. To set up the notation we will work with vectors $x \in \mathbb{R}^{\binom{n}{2}}$ that associate a weight to each edge. We index the vectors as $x_{u,v}$ where $u \neq v \in [n]$ correspond to vertices.

A result of Edmonds (1965) states that the perfect matching polytope, which is the convex hull of all edge-indicator vectors corresponding to perfect matchings, can be explicitly written in terms of "odd-cut" constraints:

$$\mathcal{P}_n := \text{conv} \left\{ \mathbf{1}_M \in \mathbb{R}^{\binom{n}{2}} \mid M \text{ is a perfect matching in } K_n \right\}$$

$$= \left\{ x \in \mathbb{R}^{\binom{n}{2}} : x \succeq 0, \forall v : \sum_u x_{u,v} = 1, \forall U \subset [n], |U| \text{ odd} \sum_{v \notin U} \sum_{u \in U} x_{u,v} \geq 1 \right\}.$$

This polytope has exponentially many vertices and exponentially many constraints. Formally, there are $V := \frac{n!}{2^{n/2}(n/2)!}$ vertices, corresponding to perfect matchings in $K_n$, and the number of constraints is $C := 2^{\Omega(n)}$ corresponding to the number of odd-sized subsets of $[n]$. By adding one dimension to account for the offsets in the inequality constraints, we can enumerate the vertices $v_1, \ldots, v_V \in \mathbb{R}^{\binom{n}{2}+1}$ and the constraints $c_1, \ldots, c_C \in \mathbb{R}^{\binom{n}{2}+1}$, such that $\langle c_i, v_j \rangle \geq 0$ for all $i, j$. Then, we define the *slack matrix* for this polytope to be $Z \in \mathbb{R}_+^{C \times V}$ with entries $Z_{i,j} = \langle c_i, v_j \rangle$.

This slack matrix clearly has rank $\binom{n}{2} + 1 = O(n^2)$. On the other hand, we claim that the non-negative rank is at least $2^{\Omega(n)}$. This follows from (a) the fact that $\mathcal{P}_n$ has extension complexity $2^{\Omega(n)}$ (Rothvoß, 2017), (b) the extension complexity of a polytope is exactly the non-negative rank of its slack matrix (Yannakakis, 1991; Fiorini et al., 2013).

Next, we define the transition operator $T$. We associate each $(x, a)$ pair with a constraint $c_i$ and each $x'$ with a vertex $v_j$. Then we define

$$T(x' \mid x, a) = \frac{\langle c_i, v_j \rangle}{\sum_{k=1}^V \langle c_i, v_k \rangle}$$

This is easily seen to be a distribution for each $(x, a)$ pair. We can represent $T$ as a $C \times V$ matrix $T = DZ$ where $D$ is a diagonal matrix (with strictly positive diagonal) and $Z$ is the slack matrix defined above.

We conclude the proof with two facts from Cohen and Rothblum (1993). First, the non-negative rank is preserved under positive diagonal rescaling, and so the non-negative rank of $T$ is also $2^{\Omega(n)}$. Second, for a row-stochastic matrix $P$, the non-negative rank is equal to the smallest number of factors we can use to write $P = RS$ where both $R$ and $S$ are row-stochastic (here factors refers to the internal dimension). It is immediate that the simplex features representation corresponds to such a row-stochastic factorization, and so we see that any simplex features representation of $T$ must have embedding dimension at least $2^{\Omega(n)}$. $\quad\square$

## A.4 On Bellman and Witness rank

We now state the formal version of Proposition 3. We consider the value-function/policy decomposition studied by Jiang et al. (2017) where we approximate the value functions with a class $\mathcal{G} : \mathcal{X} \to [0, H]$ and the policies with a class $\Pi : \mathcal{X} \to \mathcal{A}$. Given an explicit (non-stationary) reward function $R_h$ with range $[0, 1]$ and the function class $\Phi$ of candidate embeddings, we define these two

classes as:

$$\Pi(\Phi) := \left\{ \pi : x_h \mapsto \underset{a \in \mathcal{A}}{\operatorname{argmax}} \left\langle \phi_h(x_h, a), \theta_h \right\rangle + R_h(x_h, a_h) : \theta_{0:H-1} \in B_d(H\sqrt{d}), \phi_{0:H-1} \in \Phi \right\},$$

$$\mathcal{G}(\Phi) := \left\{ g : x_h \mapsto \underset{a}{\max} \left\langle \phi_h(x_h, a), \theta_h \right\rangle + R_h(x_h, a_h) : \theta_{0:H-1} \in B_d(H\sqrt{d}), \phi_{0:H-1} \in \Phi \right\}.$$

Here $B_d(\cdot)$ is the Euclidean ball in $d$ dimensions with the specified radius. We have the following proposition:

**Proposition 5.** *The low rank MDP model with any function classes $\mathcal{G} \subset \mathcal{X} \to [0, B]$ and $\Pi \subset \mathcal{X} \to \Delta(\mathcal{A})$ has bellman rank at most $d$ with normalization parameter $O(B\sqrt{d})$. Additionally, for any known reward function $R$ with range $[0, 1]$ and assuming $\phi_{0:H-1}^\star \in \Phi$, the optimal policy and value function lie in $(\mathcal{G}(\Phi), \Pi(\Phi))$, and so OLIVE has sample complexity $\tilde{O}\left(\operatorname{poly}(d, H, K, \log|\Phi|, \epsilon^{-1})\right)$.*

*Proof of Proposition 5.* The result is essentially Proposition 9 in Jiang et al. (2017), who address the simplex representation case. We address the general case and also verify the realizability assumption.

Consider any explicitly specified non-stationary reward function $R_h : \mathcal{X} \times \mathcal{A} \to [0, 1]$ and any low rank MDP with embedding functions $\phi_{0:H-1}^\star, \mu_{0:H-1}^\star$ and embedding dimension $d$. For any policies $\pi, \pi'$ and any value function $g : \mathcal{X} \to \mathbb{R}$ we define the *average Bellman error* (Jiang et al., 2017) as

$$\mathcal{E}(\pi, (g, \pi'), h) := \mathbb{E}\left[ g(x_h) - R_h(x_h, a_h) - g(x_{h+1}) \mid a_{0:h-1} \sim \pi, a_h = \pi'(x_h), \mathcal{M} \right],$$

We also introduce the shorthand

$$\Delta((g, \pi'), x_h) := \mathbb{E}\left[ g(x_h) - R_h(x_h, a_h) - g(x_{h+1}) \mid x_h, a_h = \pi'(x_h) \right].$$

Then, in the low rank MDP, the average Bellman error admits a factorization as follows

$$\begin{aligned}
\mathcal{E}(\pi, (g, \pi'), h) &= \mathbb{E}\left[ \Delta((g, \pi'), x_h) \mid x_h \sim \pi \right] \\
&= \left\langle \mathbb{E}\left[ \phi_{h-1}^\star(x_{h-1}, a_{h-1}) \mid \pi \right], \int \mu_{h-1}^\star(x_h) \Delta((g, \pi'), x_h) d(x_h) \right\rangle \\
&=: \left\langle \nu_h(\pi), \xi_h((g, \pi')) \right\rangle
\end{aligned}$$

We also have the normalization $\|\nu_h(\pi)\|_2 \leq 1$ and $\|\xi_h((g, \pi))\|_2 \leq (2B+1)\sqrt{d}$. This final calculation is based on the triangle inequality, the bounds on $g$ and $R$ and the normalization of $\mu_{h-1}^\star$. Thus for any low rank MDP and *any* (bounded) function class $\mathcal{G}, \Pi$, the Bellman rank is at most $d$ with norm parameter $O(B\sqrt{d})$.

To prove that OLIVE has low sample complexity, we need to verify that the optimal policy and optimal value function lie in $\Pi(\Phi)$ and $\mathcal{G}(\Pi)$ respectively. Then we must calculate the statistical complexity of these two classes. Observe that we can express the Bellman backup of any function $V : \mathcal{X} \to \mathbb{R}$ as a linear function in the optimal embedding $\phi^\star$:

$$\begin{aligned}
(\mathcal{T}_h V)(x, a) &:= \mathbb{E}[R_h(x, a) + V(x') \mid x, a, h] = R_h(x, a) + \left\langle \phi_h^\star(x, a), \int \mu_h^\star(x') V(x') d(x') \right\rangle \\
&= R_h(x, a) + \left\langle \phi_h^\star(x, a), w \right\rangle.
\end{aligned}$$

for some vector $w$. Moreover, if $V : \mathcal{X} \to [0, H]$, we know that $\|w\| \leq H\sqrt{d}$. In particular, this implies that the optimal $Q$ function is a linear function in the true embedding functions $\phi_{0:H-1}^\star$, and so realizability holds for $\mathcal{G}(\Phi), \Pi(\Phi)$. These function classes have range $B = O(H\sqrt{d})$ so the normalization parameter in the Bellman rank definition is $O(Hd)$.

Finally, we must calculate the statistical complexity of these two classes. For $\Pi(\Phi)$ the Natarajan dimension is at most $\tilde{O}\left(H(d + \log|\Phi|)\right)$, since for each $h$, we choose $\phi_h$ and a $d$-dimensional linear classifier. Analogously the pseudo-dimension of $\mathcal{G}(\Phi)$ is $\tilde{O}\left(H(d + \log|\Phi|)\right)$. Formally, we give a crude upper bound on the growth function, focusing on $\Pi(\Phi)$. Fix $h$, let $S$ be a sample of $n$ pairs $(x, a)$, and let $h_1, h_2 : S \to \{0, 1\}$ such that $h_1(x, a) \neq h_2(x, a)$ for all points in the sample. Since once we fix $\phi \in \Phi$, we have a linear class, we can vary $\theta$ to match $h_1, h_2$ on at most $(n+1)^d$ subsets $T \subset S$. Then by varying $\phi \in \Phi$ we can match $h_1, h_1$ in total on $|\Phi|(n+1)^d \leq n^{O(d+\log|\Phi|)}$ subsets. If $S$ is shattered, this means that $2^n \leq n^{O(d+\log|\Phi|)}$, which means that the Natarajan dimension is $O((d + \log|\Phi|)\log(d + \log|\Phi|))$. This calculation is for a fixed $h$, but the same argument yields the bound of $\tilde{O}(H(d + \log|\Phi|))$. Instantiating, we obtain the sample complexity bound for OLIVE. $\square$

For the model-based version using the witness rank, the arguments are more straightforward.

**Proposition 6.** *The low rank MDP model with any candidate model class $\mathcal{P}$ has witness rank at most $d$, with norm parameter $O(\sqrt{d})$. Additionally, for any explicitly specified reward function $R$ with range $[0, 1]$ and under Assumption 1, the algorithm of Sun et al. (2019) (with witness class of all bounded functions) has sample complexity $\tilde{O}\left(\mathrm{poly}(d, K, H, \log |\Phi||\Upsilon|, \varepsilon^{-1})\right)$.*

*Proof.* Given a model $M$ and an explicit reward function $R$, we use $\pi_M$ to denote the optimal policy for $R$ with transitions governed by $M$. Then, for two models $M_1, M_2$ and a time step $h$ the witness model misfit, when instantiated with the test function class as all bounded functions, is defined as

$$\mathcal{W}(M_1, M_2, h) := \mathbb{E}\left[\|M_2(\cdot \mid x_h, a_h) - T(\cdot \mid x_h, a_h)\|_{\mathrm{TV}} \mid a_{0:h-1} \sim \pi_{M_1}, a_h = \pi_{M_2}, \mathcal{M}\right].$$

Here we use the notation $M(\cdot \mid x_h, a_h)$ to denote the transition operator implied by $M$ at stage $h$. Recall that $T$ is the transition operator of the true MDP. In words, the witness model misfit is the one-step TV error between candidate model $M_2$ and the true environment $T$ on the data distribution induced by executing policy $\pi_{M_1}$ for $h$ steps.

Using the backing up argument from the proof of Proposition 5, it is easy to see that the witness model misfit admits a factorization as

$$\mathcal{W}(M_1, M_2, h) = \left\langle \mathbb{E}\left[\phi_{h-1}^\star(x_{h-1}, a_{h-1}) \mid \pi_{M_1}, \mathcal{M}\right], \int \mu_{h-1}^\star(x_h)\Delta(x_h, M_2) \right\rangle$$

where $\Delta(x_h, M_2)$ is the expected total variation distance between $M_2$ and $T$ on $(x_h, \pi_{M_2}(x_h))$. Based on this calculation, the witness rank is at most $d$ and the normalization parameter is at most $O(\sqrt{d})$. It is more straightforward to see that realizability holds here, and so the algorithm of Sun et al. (2019) has the stated sample complexity. $\qquad\square$

# B   Analysis of FLAMBE

As a reminder, FLAMBE interacts with a low rank MDP $\mathcal{M}$, with time horizon $H$ and with non-stationary dynamics $T_h(x_{h+1} \mid x_h, a_h) = \langle \phi_h^\star(x_h, a_h), \mu_h^\star(x_{h+1}) \rangle$. We assume that for each $h$ the operators $\phi_h^\star, \mu_h^\star$ embed into $\mathbb{R}^d$. We use the shorthand $\mathbb{E}_\pi[\cdot] = \mathbb{E}[\cdot \mid \pi, \mathcal{M}]$ to denote expectations when policy $\pi$ interacts with the real MDP $\mathcal{M}$ and $\widehat{\mathbb{E}}_\pi[\cdot] = \mathbb{E}\left[\cdot \mid \pi, \widehat{\mathcal{M}}\right]$ for expectations when the policy interacts with the estimated MDP $\widehat{\mathcal{M}}$, which has dynamics $\widehat{T}_{0:H-1}$. Note that this MDP model changes from iteration to iteration. When necessary we will use $\widehat{\mathbb{E}}_{j,\pi}[\cdot]$ to denote the MDP model learned in the $j^{\text{th}}$ iteration of FLAMBE.

The analysis of FLAMBE is based on a potential function argument. The key quantities are the second moment matrices of the real features induced by the policies $\rho_0, \rho_1, \ldots$ at each time $h$. Formally, for $h \in \{0, \ldots, H-1\}$ and $j \in [J_{\max}]$ we define

$$\Sigma_{h,j} := \lambda I_{d \times d} + \sum_{i=0}^{j-1} \mathbb{E}_{\rho_i}\left[\phi_h^\star(x_h, a_h)\phi_h^\star(x_h, a_h)^\top\right],$$

where $\lambda > 0$ is a small constant we will set towards the end of the proof. Note that $\Sigma_{h,j} \succ 0$ for all $h, j$ and that this is a cumulative (over $j$) second-moment matrix.

The importance of $\Sigma_{h,j}$ is demonstrated in the next result, which establishes an accuracy guarantee for the model $\widehat{T}_{0:H-1}$ learned in iteration $j$. The result is a corollary of Theorem 18.

**Corollary 4.** *Fix $j \geq 1$, $h \in \{1, \ldots H-1\}$, $\delta \in (0, 1)$, and let $\rho_0, \ldots, \rho_{j-1}$ be any (possibly data-dependent) policies, with $\Sigma_{h,j}$ defined accordingly. Let $D_h$ be a dataset of $nj$ examples where for each $0 \leq i < j$ we collect $n$ triples $(x_h, a_h, x_{h+1})$ by rolling in with $\rho_i$ to $x_h$ and taking $a_h$ uniformly at random. Then with probability $1 - \delta$ the output $(\widehat{\phi}_h, \widehat{\mu}_h)$ of $\mathrm{MLE}(D_h)$ satisfies*

$$\left\|\int \mu_{h-1}(x_h)\mathrm{unif}(a_h)\left\|\left\langle \widehat{\phi}_h(x_h, a_h), \widehat{\mu}_h(\cdot)\right\rangle - T_h(\cdot \mid x_h, a_h)\right\|_{\mathrm{TV}}\right\|_{\Sigma_{h-1,j}}^2 \leq \lambda d + \frac{2\log(|\Phi||\Upsilon|/\delta)}{n}.$$

*Additionally, for any $j \geq 1$, with probability at least $1 - \delta$ we have*

$$\left\|\left\langle \widehat{\phi}_0(x_0, a_0), \widehat{\mu}_0(\cdot)\right\rangle - T(\cdot \mid x_0, a_0)\right\|_{\mathrm{TV}}^2 \leq \frac{2\log(|\Phi||\Upsilon|/\delta)}{n}.$$

*Proof.* For shorthand, we use $v_h$ to denote the $d$-dimensional vector on the left hand side of the desired bound. Then, the left hand side is

$$\|v_h\|_{\Sigma_{h-1,j}}^2 = \lambda \|v_h\|_2^2 + \sum_{i=0}^{j-1} \mathbb{E}_{\rho_i} \left[ \left( \phi_{h-1}^\star (x_{h-1}, a_{h-1})^\top v_h \right)^2 \right]$$

$$= \lambda \|v_h\|_2^2 + \sum_{i=0}^{j-1} \mathbb{E}_{\rho_i} \left[ \left( \mathbb{E} \left[ \left\| \left\langle \widehat{\phi}_h(x_h, a_h), \widehat{\mu}_h(\cdot) \right\rangle - T_h(\cdot \mid x_h, a_h) \right\|_{\mathrm{TV}} \mid x_{h-1}, a_{h-1} \right] \right)^2 \right]$$

$$\leq \lambda d + \sum_{i=0}^{j-1} \mathbb{E}_{\rho_i} \left[ \left\| \left\langle \widehat{\phi}_h(x_h, a_h), \widehat{\mu}_h(\cdot) \right\rangle - T_h(\cdot \mid x_h, a_h) \right\|_{\mathrm{TV}}^2 \mid a_h \sim \mathrm{unif}(\mathcal{A}) \right].$$

The first term appears in the desired bound, so now we focus on the second term. We have $nj$ total examples that form a martingale process, since $\rho_i$ depends on all of the data collected in previous iterations. Applying Theorem 18, we see that with probability $1 - \delta$:

$$\sum_{i=0}^{j-1} n \cdot \mathbb{E}_{\rho_i} \left[ \left\| \left\langle \widehat{\phi}_h(x_h, a_h), \widehat{\mu}_h(\cdot) \right\rangle - T_h(\cdot \mid x_h, a_h) \right\|_{\mathrm{TV}}^2 \mid a_h \sim \mathrm{unif}(\mathcal{A}) \right] \leq 2 \log(|\Phi||\Upsilon/\delta),$$

where the factor of $n$ arises since we collect $n$ examples from $\rho_i$. Re-arranging we obtain the first bound. The bound for $h = 0$ is a direct application of Theorem 18, since we assume there is a fixed starting state $x_0$ with a single available action. $\qquad \square$

Now that we have established an accuracy guarantee in terms of the previous exploratory policies, we state and prove the main technical "simulation" lemma. The following notation is helpful. Given an MDP model $\widehat{\phi}_{0:H-1}, \widehat{\mu}_{0:H-1}$ and positive definite matrices $\Sigma_0, \ldots, \Sigma_{H-1}$, define

$$\forall h \geq 1 : \mathrm{err}_h(\Sigma_{h-1}) := \left\| \int \mu_{h-1}(x_h) \mathrm{unif}(a_h) \left\| \left\langle \widehat{\phi}_h(x_h, a_h), \widehat{\mu}_h(\cdot) \right\rangle - T_h(\cdot \mid x_h, a_h) \right\|_{\mathrm{TV}} \right\|_{\Sigma_{h-1}}^2,$$

$$\mathrm{err}_0 := \left\| \left\langle \widehat{\phi}_0(x_0, a_0), \widehat{\mu}_0(\cdot) \right\rangle - T_0(\cdot \mid x_0, a_0) \right\|_{\mathrm{TV}}^2.$$

Further, for each $h \geq 1$, define $\mathcal{K}_h(\Sigma_h) := \left\{ (x, a) \in \mathcal{X} \times \mathcal{A} : \|\phi_h^\star(x_h, a_h)\|_{\Sigma_h^{-1}}^2 \leq 1 \right\}$. Let $M_{\mathcal{K}}$ be the MDP with non-stationary transition operator $T_{h,\mathcal{K}}$ defined as

$$T_{h,\mathcal{K}}(x_{h+1} \mid x_h, a_h) = \begin{cases} \langle \phi_h^\star(x_h, a_h), \mu_h^\star(x_{h+1}) \rangle & \text{if } (x_h, a_h) \in \mathcal{K}_h(\Sigma_h) \\ \mathbf{1}\{x_{h+1} = x_{\mathrm{absorb}}\} & \text{if } (x_h, a_h) \notin \mathcal{K}_h(\Sigma_h) \end{cases},$$

where $x_{\mathrm{absorb}}$ is a special self-looping absorbing state with a single action $a_{\mathrm{absorb}}$ such that $T(x_{\mathrm{absorb}} \mid x_{\mathrm{absorb}}, a_{\mathrm{absorb}}) = 1$ always. The initial transition $T_{0,\mathcal{K}}$ is identical to $T_0$. The intuition is that $\mathcal{K}$ denotes the set of "known" state-action pairs, and the MDP $M_{\mathcal{K}}$ terminates any episode that escapes the known set. In all of these definitions, we suppress the dependence on $\Sigma_h$ when it is clear from context. We always consider $(x_{\mathrm{absorb}}, a_{\mathrm{absorb}})$ to be *known*.

**Lemma 5.** *Let* $\widehat{\phi}_{0:H-1}, \widehat{\mu}_{0:H-1}$ *be an MDP model and let* $\Sigma_{0:H-1}$ *be positive definite matrices. Assume that*

$$\forall h \in \{0, \ldots, H-1\} : \ \mathrm{err}_h(\Sigma_{h-1}) \leq \varepsilon_{\mathrm{TV}}.$$

*Let* $f : \mathcal{X} \times \mathcal{A} \to [0, 1]$ *be any function such that* $f(x_{\mathrm{absorb}}, a_{\mathrm{absorb}}) = 0$, *and let* $\pi$ *be any policy. Then for any* $h \in \{0, \ldots, H-1\}$

$$\mathbb{E}_\pi \left[ f(x_h, a_h) \mid M_{\mathcal{K}} \right] - HK\sqrt{\varepsilon_{\mathrm{TV}}} \leq \widehat{\mathbb{E}}_\pi \left[ f(x_h, a_h) \right] \leq \mathbb{E}_\pi \left[ f(x_h, a_h) \mid M_{\mathcal{K}} \right] + HK\sqrt{\varepsilon_{\mathrm{TV}}}$$

$$+ \sum_{h'=0}^{h-1} \mathbb{P}\left[ (x_{h'}, a_{h'}) \notin \mathcal{K}_{h'} \mid \pi, M_{\mathcal{K}} \right].$$

This lemma establishes a sharp relationship between the learned MDP $\widehat{\mathcal{M}}$ and an *absorbing* MDP $\mathcal{M}_{\mathcal{K}}$, defined in terms of the matrices $\Sigma_h$, which also governs the estimation error for $\widehat{\mathcal{M}}$. Intuitively

the error guarantee implies that $\widehat{\mathcal{M}}$ closely approximates $\mathcal{M}_\mathcal{K}$ provided we stay within the known set $\mathcal{K}_{0:H-1}$. Conversely the difference in value between the two MDPs can be bounded in terms of the escaping probability, which is the third term on the right hand side of the bound.

Note also that the above lemma, with $\varepsilon_{\mathrm{TV}} = 0$, can be used to compare $\mathcal{M}$ with $\mathcal{M}_\mathcal{K}$, which yields that for any non-negative function $f$ and any policy $\pi$:

$$\mathbb{E}_\pi \left[ f(x_h, a_h) \mid \mathcal{M}_\mathcal{K} \right] \le \mathbb{E}_\pi \left[ f(x_h, a_h) \right] \le \mathbb{E}_\pi \left[ f(x_h, a_h) \mid \mathcal{M}_\mathcal{K} \right] + \sum_{h=0}^{h-1} \mathbb{P}\left[ (x_{h'}, a_{h'}) \notin \mathcal{K}_{h'} \mid \pi, \mathcal{M}_\mathcal{K} \right].$$

(3)

This bound actually holds for any sets $\mathcal{K}_h$. We now turn to the proof of Lemma 5.

*Proof.* Let $\widehat{V}_{h'}(x) := \widehat{\mathbb{E}}_\pi[f(x_h, a_h) \mid x_{h'} = x]$ denote the value function (relative to $f$) in the model and let $V_{h',\mathcal{K}}(x)$ denote the analogous quantity in the absorbing MDP. We have the following telescoping identity:

$$\widehat{\mathbb{E}}_\pi[f(x_h, a_h)] - \mathbb{E}_\pi \left[ f(x_h, a_h) \mid M_\mathcal{K} \right] = \int \left( \widehat{T}_0(x_1 \mid x_0, a_0) - T_{0,\mathcal{K}}(x_1 \mid x_0, a_0) \right) \widehat{V}_1(x_1)$$
$$+ \mathbb{E}_\pi \left[ \widehat{V}_1(x_1) - V_{1,\mathcal{K}}(x_1) \mid M_\mathcal{K} \right]$$
$$= \sum_{h'=0}^{h-1} \mathbb{E}_\pi \left[ \int (\widehat{T}_{h'}(x_{h'+1} \mid x_{h'}, a_{h'}) - T_{h',\mathcal{K}}(x_{h'+1} \mid x_{h'}, a_{h'})) \widehat{V}_{h'+1}(x_{h'+1}) \mid M_\mathcal{K} \right].$$

Note that $\widehat{V}_h(x) = V_{h,\mathcal{K}}(x)$ at the time $h$ where we apply function $f$. This means that we only accumulate errors up to time $h - 1$. We now work with one of these terms. In the remainder of the proof, unless otherwise specified, all expectations are taken by executing $\pi$ in $M_\mathcal{K}$. By adding and subtracting $T_{h'}$, we get two terms

$$\mathrm{Term1}_{h'} := \mathbb{E} \left[ \int (\widehat{T}_{h'}(x_{h'+1} \mid x_{h'}, a_{h'}) - T_{h'}(x_{h'+1} \mid x_{h'}, a_{h'})) \widehat{V}_{h'+1}(x_{h'+1}) \right]$$
$$\mathrm{Term2}_{h'} := \mathbb{E} \left[ \int (T_{h'}(x_{h'+1} \mid x_{h'}, a_{h'}) - T_{h',\mathcal{K}}(x_{h'+1} \mid x_{h'}, a_{h'})) \widehat{V}_{h'+1}(x_{h'+1}) \right].$$

For $\mathrm{Term1}_{h'}$, note that the expression evaluates to zero if $x_{h'} = x_{\mathrm{absorb}}$ since both $\widehat{T}_{h'}$ and $T_{h'}$ agree that $x_{\mathrm{absorb}}$ has a single self-looping action. We now bound Term1 at time $h' = 1$, although exactly the same argument applies to $h' > 0$. Defining $\mathrm{err}(x_1, a_1) := \left\| \widehat{T}_1(\cdot \mid x_1, a_1) - T_1(\cdot \mid x_1, a_1) \right\|_{\mathrm{TV}}$ and by applying Holder's inequality, we have

$$\mathrm{Term1}_1 \le \mathbb{E} \left[ \mathbf{1}\{x_1 \ne x_{\mathrm{absorb}}\} \left\| \widehat{T}_1(\cdot \mid x_1, a_1) - T_1(\cdot \mid x_1, a_1) \right\|_{\mathrm{TV}} \right]$$
$$\le K \cdot \mathbb{E} \left[ \mathbf{1}\{x_1 \ne x_{\mathrm{absorb}}\} \mathrm{unif}(a_1) \mathrm{err}(x_1, a_1) \right]$$
$$= K \cdot \mathbb{E} \left[ \phi_0^\star(x_0, a_0) \mathbf{1} \left\{ \|\phi_0^\star(x_0, a_0)\|_{\Sigma_0^{-1}}^2 \le 1 \right\} \right] \cdot \int \mu_0^\star(x_1) \mathrm{unif}(a_1) \mathrm{err}(x_1, a_1)$$
$$\le K \cdot \mathbb{E} \left[ \|\phi_0^\star(x_0, a_0)\|_{\Sigma_0^{-1}} \mathbf{1} \left\{ \|\phi_0^\star(x_0, a_0)\|_{\Sigma_0^{-1}}^2 \le 1 \right\} \right] \cdot \sqrt{\mathrm{err}_1(\Sigma_0)}$$
$$\le K\sqrt{\varepsilon_{\mathrm{TV}}}$$

The first inequality is Holder's inequality, while the second is an importance weighting argument to replace $a_1 \sim \pi(x_1)$ with the uniform distribution. Next we re-write the expectation using the low rank dynamics, and also use the fact that $x_1 \ne x_{\mathrm{absorb}}$ implies that the previous transition was non-absorbing, which yields the indicator. Finally, we use the Cauchy-Schwarz inequality in the $\Sigma_0$ norm, along with the implication of the indicator and the assumed bound on $\mathrm{err}_1(\Sigma_0)$. This argument applies as is to all indices $h' > 0$ and for $h' = 0$ we simply apply Holder's inequality and the definition of $\mathrm{err}_0$ to obtain the upper bound $\sqrt{\varepsilon_{\mathrm{TV}}}$. In total, these terms account for the $HK\sqrt{\varepsilon_{\mathrm{TV}}}$ terms on both sides of the lemma statement.

Next we turn to $\text{Term}2_{h'}$. For the first inequality in the lemma statement, we need to upper bound $-\text{Term}2_{h'}$, but this term is easily seen to be non-positive, since $\widehat{V}(x_{\text{absorb}}) = 0$ always. So this proves the first inequality. For the second inequality, we have (again focusing on time 1)

$$\text{Term}2_1 = \mathbb{E}\left[\int T_1(x_2 \mid x_1, a_1)\mathbf{1}\{(x_1, a_1) \notin \mathcal{K}_1\}\widehat{V}_2(x_2)\right] \leq \mathbb{P}\left[(x_1, a_1) \notin \mathcal{K}_1 \mid \pi, M_{\mathcal{K}}\right].$$

The same argument applies for all $h \geq 1$. $\qquad\square$

In the next lemma, we consider the case where $\rho_j$ has large escaping probability, measured with respect to the known sets $\mathcal{K}_h(\Sigma_{h,j})$. Recall that $\Sigma_{h,j}$ is the second moment matrix of the true features $\phi_h^\star$ at time $h$ induced by the previous roll-in policies $\rho_0, \ldots, \rho_{j-1}$.

**Lemma 6.** *Consider iteration $j$ of* FLAMBE *and assume that* $\text{err}_h(\Sigma_{h-1,j}) \leq \varepsilon_{\text{TV}}$ *for each $h$ with our current model $\widehat{\mathcal{M}}$. Define $R_h(x, a) := \mathbf{1}\{(x, a) \notin \mathcal{K}_h(\Sigma_{h,j})\}$ for each $h$. Then,*

$$\max_h \text{tr}\left(\mathbb{E}_{\rho_j}\left[\phi_h^\star(x_h, a_h)\phi_h^\star(x_h, a_h)\right]\Sigma_{h,j-1}^{-1}\right) \geq \frac{1}{H}\max_h\left\{\widehat{\mathbb{E}}_{\rho_j}\left[R_h(x_h, a_h)\right] - HK\sqrt{\varepsilon_{\text{TV}}}\right\}$$

*Proof.* For shorthand let $\mathcal{K}_h := \mathcal{K}_h(\Sigma_{h,j})$ denote the known set at round $j$, and let $\mathcal{M}_{\mathcal{K}}$ denote the corresponding absorbing MDP. Then, applying the second inequality in Lemma 5 we have

$$\widehat{\mathbb{E}}_{\rho_j}\left[R_h(x_h, a_h)\right] \leq \mathbb{E}\left[R_h(x_h, a_h) \mid \rho_j, \mathcal{M}_{\mathcal{K}}\right] + HK\sqrt{\varepsilon_{\text{TV}}} + \sum_{h'=0}^{h-1}\mathbb{P}\left[(x_{h'}, a_{h'}) \notin \mathcal{K}_{h'} \mid \rho_j, \mathcal{M}_{\mathcal{K}}\right]$$

$$\leq HK\sqrt{\varepsilon_{\text{TV}}} + \sum_{h'=0}^{h}\mathbb{P}\left[(x_{h'}, a_{h'}) \notin \mathcal{K}_{h'} \mid \rho_j, \mathcal{M}_{\mathcal{K}}\right]$$

$$\leq HK\sqrt{\varepsilon_{\text{TV}}} + \sum_{h'=0}^{h}\mathbb{P}\left[(x_{h'}, a_{h'}) \notin \mathcal{K}_{h'} \mid \rho_j, \mathcal{M}\right]$$

$$= HK\sqrt{\varepsilon_{\text{TV}}} + \sum_{h'=0}^{h}\mathbb{P}\left[\|\phi_{h'}^\star(x_{h'}, a_{h'})\|_{\Sigma_{h',j}^{-1}} \geq 1 \mid \rho_j, \mathcal{M}\right]$$

$$\leq HK\sqrt{\varepsilon_{\text{TV}}} + \sum_{h'=0}^{h}\mathbb{E}_{\rho_j}\left[\text{tr}\left(\phi_{h'}^\star(x_{h'}, a_{h'})\phi_{h'}^\star(x_{h'}, a_{h'})^\top\Sigma_{h',j}^{-1}\right)\right]$$

The last step follows from Markov's inequality. Since both matrices are positive semidefinite, the trace terms are all non-negative. Therefore, by the pigeonhole principle, there exists some $h' \in \{0, \ldots, h\}$ for which

$$\mathbb{E}_{\rho_j}\left[\text{tr}\left(\phi_{h'}^\star(x_{h'}, a_{h'})\phi_{h'}^\star(x_{h'}, a_{h'})^\top\Sigma_{h',j}^{-1}\right)\right] \geq \frac{1}{H}\left(\widehat{\mathbb{E}}_{\rho_j}\left[R_h(x_h, a_h)\right] - HK\sqrt{\varepsilon_{\text{TV}}}\right).$$

This argument applies for all $R_h$, and so we obtain the lemma. $\qquad\square$

Next we argue that there cannot be too many iterations for which $\max_h \widehat{\mathbb{E}}_{\rho_j}\left[R_h(x_h, a_h)\right]$ is large. For notation, here we use $\widehat{\mathcal{M}}^{(j)}$ to denote the MDP model in iteration $j$ and we use $R_h^{(j)}$ to denote the reward functions in Lemma 6 derived from the known sets in iteration $j$.

**Corollary 7.** *Assume that for each round $j \in [J_{\max}]$ and for all $h$ we have $\text{err}_h(\Sigma_{h-1,j}) \leq \varepsilon_{\text{TV}}$. Set*

$$J_{\max} := \frac{4Hd}{\lambda K\sqrt{\varepsilon_{\text{TV}}}} \cdot \log\left(1 + \frac{4H}{\lambda K\sqrt{\varepsilon_{\text{TV}}}}\right). \tag{4}$$

*Then there exists some $j \in [J_{\max}]$ for which $\max_h \mathbb{E}\left[R_h^{(j)}(x_h, a_h) \mid \rho_j, \widehat{\mathcal{M}}^{(j)}\right] \leq 2HK\sqrt{\varepsilon_{\text{TV}}}$.*

*Proof.* Suppose that in round $j$, it holds that $\max_h \mathbb{E}\left[R_h^{(j)}(x_h, a_h) \mid \rho_j, \widehat{\mathcal{M}}^{(j)}\right] \geq 2HK\sqrt{\varepsilon_{\mathrm{TV}}}$. Then, by Lemma 6, there exists some time step $h$ for which

$$\mathrm{tr}\left(\mathbb{E}_{\rho_j}\left[\phi_h^\star(x_h, a_h)\phi_h^\star(x_h, a_h)^\top\right]\Sigma_{h,j}^{-1}\right) \geq K\sqrt{\varepsilon_{\mathrm{TV}}}.$$

Note that we also have $\Sigma_{h,j+1} = \Sigma_{h,j} + \mathbb{E}_{\rho_j}\left[\phi_h^\star(x_h, a_h)\phi_h^\star(x_h, a_h)^\top\right]$, so we are in a position to apply the elliptical potential argument. Specifically if $J$ is the number of iterations for which the above inequality holds for some $h$, then applying Lemma 23 for each $h$ and summing across $h$ yields

$$JK\sqrt{\varepsilon_{\mathrm{TV}}} \leq (1 + 1/\lambda)dH\log(1 + J_{\max}/d)$$

Plugging in our choice of $J_{\max}$, and using the fact that $\lambda < 1$ we have

$$J < \frac{2Hd}{\lambda K\sqrt{\varepsilon_{\mathrm{TV}}}}\log\left(1 + \frac{4H}{\lambda K\sqrt{\varepsilon_{\mathrm{TV}}}}\log\left(1 + \frac{4H}{\lambda K\sqrt{\varepsilon_{\mathrm{TV}}}}\right)\right)$$

$$\leq \frac{2Hd}{\lambda K\sqrt{\varepsilon_{\mathrm{TV}}}}\log\left(1 + \left(\frac{4H}{\lambda K\sqrt{\varepsilon_{\mathrm{TV}}}}\right)^2\right) \leq J_{\max}.$$

This means that in $J_{\max}$ iterations, we can have $\max_h \mathbb{E}\left[R_h^{(j)}(x_h, a_h) \mid \rho_j, \widehat{\mathcal{M}}^{(j)}\right] \geq 2HK\sqrt{\varepsilon_{\mathrm{TV}}}$ in at most $J < J_{\max}$ of them. Thus we must have one where this quantity is small, which proves the lemma. $\square$

Next, we state a guarantee provided by Algorithm 2, which is a more convenient form of Lemma 16.

**Lemma 8.** *Fix any iteration $j$, time $h$, function $f : \mathcal{X} \times \mathcal{A} \to [0, 1]$, policy $\pi$, any $\alpha > 0$. Then*

$$\mathbb{E}\left[f(x_h, a_h) \mid \pi, \widehat{\mathcal{M}}^{(j)}\right] \leq \frac{T\beta}{2\alpha} + \frac{\alpha d}{2T} + \frac{\alpha KH}{2}\mathbb{E}\left[f(x_h, a_h) \mid \rho_j, \widehat{\mathcal{M}}^{(j)}\right],$$

*where $T \leq 4d\log(1 + 4/\beta)/\beta$ and $\beta > 0$ is the parameter to Algorithm 2.*

*Proof.* We suppress the dependence on $j$. Let us first focus on $\rho_h^{\mathrm{pre}}$, which is output of Algorithm 2 for some time step $h$. $\rho_h^{\mathrm{pre}}$ induces a distribution over states at time step $h$, and we argue that this distribution adequately covers all possible roll-in distributions in the model $\widehat{\mathcal{M}} = \widehat{\mathcal{M}}^{(j)}$. Consider any function $f : \mathcal{X} \times \mathcal{A} \to [0, 1]$, any policy $\pi$, any $\Sigma \succ 0$, and $\alpha > 0$. Calling $f_\pi(x_h) = \int \pi(a_h \mid x_h)f(x_h, a_h)$, we have

$$\widehat{\mathbb{E}}_\pi f(x_h, a_h) = \widehat{\mathbb{E}}_\pi \left\langle \widehat{\phi}_{h-1}(x_{h-1}, a_{h-1}), \int \widehat{\mu}_{h-1}(x_h)f_\pi(x_h) \right\rangle$$

$$\leq \widehat{\mathbb{E}}_\pi \left\|\widehat{\phi}_{h-1}(x_{h-1}, a_{h-1})\right\|_{\Sigma^{-1}} \cdot \left\|\int \widehat{\mu}_{h-1}(x_h)f_\pi(x_h)\right\|_\Sigma$$

$$\leq \frac{1}{2\alpha}\widehat{\mathbb{E}}_\pi \left\|\phi_{h-1}(x_{h-1}, a_{h-1})\right\|_{\Sigma^{-1}}^2 + \frac{\alpha}{2}\left\|\int \widehat{\mu}_{h-1}(x_h)f_\pi(x_h)\right\|_\Sigma^2.$$

Here we expand $\widehat{T}_{h-1}$ in terms of its low rank representation and then apply the Cauchy-Schwarz inequality in the norm induced by $\Sigma$. Finally we use the AM-GM inequality which holds for any non-negative $\alpha$.

We instantiate $\Sigma$ to be the covariance matrix induced by $\rho_h^{\mathrm{pre}}$. First, for any policy $\pi$ we define the $h - 1$ step model covariance as $\Sigma_\pi := \widehat{\mathbb{E}}_\pi\widehat{\phi}_{h-1}(x_{h-1}, a_{h-1})\widehat{\phi}_{h-1}(x_{h-1}, a_{h-1})^\top$, where the dependence on $h - 1$ is suppressed in the notation. Note that both the expectation and the embedding are taken with respect to the model $\widehat{\mathcal{M}}$. Then, the output of Algorithm 2 is a $h$-step policy $\rho_h^{\mathrm{pre}}$ that is defined as a mixture over $T$ policies $\pi_1, \ldots, \pi_T$. Using these policies, we define $\Sigma$ as follows:

$$\Sigma = \Sigma_{\rho_h^{\mathrm{pre}}} + \frac{I_{d\times d}}{T} = \frac{1}{T}\sum_{t=1}^{T}\Sigma_{\pi_t} + \frac{I_{d\times d}}{T}.$$

As we run [Algorithm 2](#) using $\widehat{T}_{0:h-1}$ we can apply [Lemma 16](#) on the $h$ step MDP $\widehat{T}_{0:h-1}$. In other words, in [Lemma 16](#), we set $H \leftarrow h$ and $\widetilde{\mathcal{M}} \leftarrow \widehat{\mathcal{M}}$. The conclusion is that $T \leq 4d \log(1 + 4/\beta)/\beta$, where $\beta$ is the parameter to the subroutine, and we can also bound the first term above:

$$\widehat{\mathbb{E}}_\pi \left\| \widehat{\phi}_{h-1}(x_{h-1}, a_{h-1}) \right\|_{\Sigma^{-1}}^2 = \widehat{\mathbb{E}}_\pi \phi_{h-1}(x_{h-1}, a_{h-1})^\top \left( \Sigma_{\rho_h^{\mathrm{pre}}} + \frac{I_{d \times d}}{T} \right)^{-1} \phi_{h-1}(x_{h-1}, a_{h-1}) \leq T\beta.$$

Next, we turn to the second term. Expanding the definition of $\Sigma$, we have

$$\left\| \int \widehat{\mu}_{h-1}(x_h) f_\pi(x_h) \right\|_\Sigma^2$$

$$= \widehat{\mathbb{E}}_{\rho_h^{\mathrm{pre}}} \left( \left\langle \widehat{\phi}_{h-1}(x_{h-1}, a_{h-1}), \int \widehat{\mu}_{h-1}(x_h) f_\pi(x_h) \right\rangle \right)^2 + \frac{\left\| \int \widehat{\mu}_{h-1}(x_h) f_\pi(x_h) \right\|_2^2}{T}$$

$$= \widehat{\mathbb{E}}_{\rho_h^{\mathrm{pre}}} \left( \widehat{\mathbb{E}} \left[ f_\pi(x_h) \mid x_{h-1}, a_{h-1} \right] \right)^2 + \frac{\left\| \int \widehat{\mu}_{h-1}(x_h) f_\pi(x_h) \right\|_2^2}{T}$$

$$\leq \widehat{\mathbb{E}}_{\rho_h^{\mathrm{pre}}} f_\pi(x_h) + \frac{\left\| \int \widehat{\mu}_{h-1}(x_h) f_\pi(x_h) \right\|_2^2}{T} \leq \widehat{\mathbb{E}}_{\rho_h^{\mathrm{pre}}} f_\pi(x_h) + d/T.$$

The first inequality is Jensen's inequality along with the fact that $f(x_h)^2 \leq f(x_h)$ since $f : \mathcal{X} \to [0, 1]$. The second inequality is based on our normalization assumptions on $\mu_{h-1}$, which we also impose on $\widehat{\mu}_{h-1}$. Finally, collecting all the terms and importance weighting the last action, we obtain the bound

$$\widehat{\mathbb{E}}_\pi f(x_h) \leq \frac{T\beta}{2\alpha} + \frac{\alpha K}{2} \widehat{\mathbb{E}}_{\rho_h^{\mathrm{pre}} \circ \mathrm{unif}(\mathcal{A})} f(x_h, a_h) + \frac{\alpha d}{2T}.$$

This bound applies to $\rho_h^{\mathrm{pre}}$. As $\rho_j$ is a uniform mixture of these policies and as $f$ is non-negative, we see that $\widehat{\mathbb{E}}_{\rho_h^{\mathrm{pre}} \circ \mathrm{unif}(\mathcal{A})} f(x_h, a_h) \leq H \cdot \widehat{\mathbb{E}}_{\rho_j} f(x_h, a_h)$, which proves the lemma. $\qquad\square$

Finally, we use the guarantee for [Algorithm 2](#), to prove that our model $\widehat{\mathcal{M}}$ universally approximate the true MDP as soon as $\max_h \mathbb{E}\left[ R_h^{(j)}(x_h, a_h) \mid \rho_j, \widehat{\mathcal{M}}^{(j)} \right] \leq 2HK\sqrt{\varepsilon_{\mathrm{TV}}}$. For the lemma, we use the concept of a sparse reward function. $R : \mathcal{X} \times \mathcal{A} \to [0, 1]$ is called *sparse* if all value functions are in $[0, 1]$. For example, this holds if $R$ is only associated with state-action pairs at a single time point.

**Lemma 9.** *Assume that for each round $j \in [J_{\max}]$ and for all $h$, we have $\mathrm{err}_h(\Sigma_{h-1,j}) \leq \varepsilon_{\mathrm{TV}}$, and set $J_{\max}$ as in [(4)](#). Then the final MDP model $\widehat{\mathcal{M}}$ satisfies the following guarantee: For any sparse reward function $R : \mathcal{X} \times \mathcal{A} \to [0, 1]$, any policy $\pi$, and any $\alpha > 0$ we have*

$$\left| V(\pi; R, \widehat{\mathcal{M}}) - V(\pi; R, \mathcal{M}) \right| \leq HK\sqrt{\varepsilon_{\mathrm{TV}}} + H\varepsilon_{\mathrm{escape}},$$

*where $\varepsilon_{\mathrm{escape}} := \alpha H^2 K^2 \sqrt{\varepsilon_{\mathrm{TV}}} + \frac{T\beta}{2\alpha} + \frac{\alpha d}{2T} + HK\sqrt{\varepsilon_{\mathrm{TV}}}$, $T \leq 4d \log(1 + 4/\beta)/\beta$ and $\beta > 0$ is the parameter to [Algorithm 2](#).*

*Proof.* Via [Corollary 7](#), there must be some round $j$ for which

$$\max_h \mathbb{P}\left[ (x_h, a_h) \notin \mathcal{K}_h(\Sigma_{h,j}) \mid \rho_j, \widehat{\mathcal{M}}^{(j)} \right] = \max_h \mathbb{E}\left[ R_h^{(j)}(x_h, a_h) \mid \rho_j, \widehat{\mathcal{M}}^{(j)} \right] \leq 2HK\sqrt{\varepsilon_{\mathrm{TV}}}.$$

We will prove the guarantee for this round $j$, and at the end of the proof argue that this also applies to the final learned model.

Combining the lower bound of the simulation lemma ([Lemma 5](#)) with the planning guarantee ([Lemma 8](#)) we see that, for any policy $\pi$

$$\mathbb{P}\left[ (x_h, a_h) \notin \mathcal{K}_h(\Sigma_{h,j}) \mid \pi, \mathcal{M}_{\mathcal{K}}^{(j)} \right] \leq \mathbb{P}\left[ (x_h, a_h) \notin \mathcal{K}_h(\Sigma_{h,j} \mid \pi, \widehat{\mathcal{M}}^{(j)} \right] + HK\sqrt{\varepsilon_{\mathrm{TV}}}$$

$$\leq \frac{\alpha KH}{2} \mathbb{P}\left[ (x_h, a_h) \notin \mathcal{K}_h(\Sigma_{h,j}) \mid \rho_j, \widehat{\mathcal{M}}^{(j)} \right] + \frac{T\beta}{2\alpha} + \frac{\alpha d}{2T} + HK\sqrt{\varepsilon_{\mathrm{TV}}}$$

$$\leq \alpha K^2 H^2 \sqrt{\varepsilon_{\mathrm{TV}}} + \frac{T\beta}{2\alpha} + \frac{\alpha d}{2T} + HK\sqrt{\varepsilon_{\mathrm{TV}}} =: \varepsilon_{\mathrm{escape}}$$

Now that we have upper bounded the escaping probability, we can turn to the approximation guarantee. While we are not in the exact setting of Lemma 5, since we have a sparse reward function, all values are in $[0,1]$ so the same argument applies. For one side of the error guarantee, since we assume that $\mathrm{err}_h(\Sigma_{h-1,j}) \le \varepsilon_{\mathrm{TV}}$ for all iterations, we have

$$V(\pi; R, \widehat{\mathcal{M}}^{(j)}) \le V(\pi; R, \mathcal{M}_{\mathcal{K}}^{(j)}) + HK\sqrt{\varepsilon_{\mathrm{TV}}} + \sum_{h=0}^{H-1} \mathbb{P}\left[ (x_h, a_h) \notin \mathcal{K}_h(\Sigma_{h,j}) \mid \pi, \mathcal{M}_{\mathcal{K}}^{(j)} \right]$$

$$\le V(\pi; R, \mathcal{M}) + HK\sqrt{\varepsilon_{\mathrm{TV}}} + \sum_{h=0}^{H-1} \mathbb{P}\left[ (x_h, a_h) \notin \mathcal{K}_h(\Sigma_{h,j}) \mid \pi, \mathcal{M}_{\mathcal{K}}^{(j)} \right]$$

$$\le V(\pi; R, \mathcal{M}) + HK\sqrt{\varepsilon_{\mathrm{TV}}} + H\varepsilon_{\mathrm{escape}}.$$

Here the first inequality is Lemma 5, while the second is due to (3). For the other direction, we first use (3) and then Lemma 5:

$$V(\pi; R, \mathcal{M}) \le V(\pi; R, \mathcal{M}_{\mathcal{K}}^{(j)}) + \sum_{h=0}^{H-1} \mathbb{P}\left[ (x_h, a_h) \notin \mathcal{K}_h(\Sigma_{h,j}) \mid \pi, \mathcal{M}_{\mathcal{K}}^{(j)} \right]$$

$$\le V(\pi; R, \widehat{\mathcal{M}}^{(j)}) + HK\sqrt{\varepsilon_{\mathrm{TV}}} + H\varepsilon_{\mathrm{escape}}.$$

This proves the result for the MDP model $\widehat{\mathcal{M}}^{(j^\star)}$ at the time $j^\star$ where the exploratory policy $\rho_{j^\star}$ fails to achieve large reward on $R_h^{(j^\star)}$. We now claim this applies for all iterations after $j^\star$, and in particular it holds at the end of the algorithm. To see why, observe that $\Sigma_{h,j+1} \succeq \Sigma_{h,j}$ for all $j$, and so $\mathcal{K}_h(\Sigma_{h,j}) \subset \mathcal{K}_h(\Sigma_{h,j+1})$ for all rounds. Since the known set increases with $j$, the escaping probability is decreasing, so it is upper bounded by $\varepsilon_{\mathrm{escape}}$ for all rounds after $j \ge j^\star$. Additionally, we assume that $\mathrm{err}_h(\Sigma_{h-1,j}) \le \varepsilon_{\mathrm{TV}}$ for all $j \in [J_{\max}]$, so the total variation term in Lemma 5 remains bounded. Working through the proof of Lemma 5, we can see that the bound continues to hold for all $j^\star \le j \le J_{\max}$, which proves the result. $\qquad\square$

**Final steps.** Let us collect all of the conditions and bounds here. At the end of the algorithm, we have

$$\max_{\pi, R} \left| V(\pi; R, \widehat{\mathcal{M}}) - V(\pi; R, \mathcal{M}) \right| \le HK\sqrt{\varepsilon_{\mathrm{TV}}} + H\varepsilon_{\mathrm{escape}}, \qquad (5)$$

where we may set

$$\varepsilon_{\mathrm{escape}} := \min_{\alpha > 0} \left\{ \alpha H^2 K^2 \sqrt{\varepsilon_{\mathrm{TV}}} + \frac{T\beta}{2\alpha} + \frac{\alpha d}{2T} + HK\sqrt{\varepsilon_{\mathrm{TV}}} \right\}, \qquad T \le \frac{4d\log(1 + 4/\beta)}{\beta},$$

and $\beta > 0$ is the parameter to Algorithm 2. Applying Corollary 4 and taking a union bound over all iterations $j \in [J_{\max}]$ and all times $h$, we can set

$$\varepsilon_{\mathrm{TV}} := \lambda d + \frac{2\log(J_{\max} H |\Phi||\Upsilon|/\delta)}{n},$$

where $\lambda > 0$ is a parameter in the analysis. Finally, the total number of samples collected is

$$nHJ_{\max}, \qquad \text{where,} \qquad J_{\max} := \frac{4Hd}{\lambda K\sqrt{\varepsilon_{\mathrm{TV}}}} \cdot \log\left(1 + \frac{4H}{\lambda K\sqrt{\varepsilon_{\mathrm{TV}}}}\right)$$

We start by optimizing for $\alpha$ in the definition of $\varepsilon_{\mathrm{escape}}$, which yields $\alpha = \sqrt{\frac{T\beta}{2(H^2 K^2 \sqrt{\varepsilon_{\mathrm{TV}}} + d/(2T))}}$. Plugging into $\varepsilon_{\mathrm{escape}}$ and using the bound on $T$, we get

$$\varepsilon_{\mathrm{escape}} \le \sqrt{2T\beta} \cdot \left( HK\varepsilon_{\mathrm{TV}}^{1/4} + \sqrt{d/(2T)} \right) + HK\sqrt{\varepsilon_{\mathrm{TV}}}$$

$$\le 2\sqrt{8d\log(1 + 4/\beta)}HK\varepsilon_{\mathrm{TV}}^{1/4} + \sqrt{d\beta}.$$

Here we are using the fact that $\varepsilon_{\mathrm{TV}} \le 1$, which is without loss of generality, since (5) is trivial when $\varepsilon_{\mathrm{TV}} \ge 1$. Now we set $\beta = H^2 K^2 \sqrt{\varepsilon_{\mathrm{TV}}}$ so that

$$\varepsilon_{\mathrm{escape}} \le 16\sqrt{d\log(1 + 4/\varepsilon_{\mathrm{TV}})}HK\varepsilon_{\mathrm{TV}}^{1/4}.$$

Thus, we may restate the final accuracy guarantee as

$$\max_{\pi, R} \left| V(\pi; R, \widehat{\mathcal{M}}) - V(\pi; R, \mathcal{M}) \right| \leq HK\sqrt{\varepsilon_{\mathrm{TV}}} + 16\sqrt{d \log(1 + {}^4\!/\!\varepsilon_{\mathrm{TV}})} H^2 K \varepsilon_{\mathrm{TV}}^{1/4}$$

$$\leq 17\sqrt{d \log(1 + {}^4\!/\!\varepsilon_{\mathrm{TV}})} H^2 K \varepsilon_{\mathrm{TV}}^{1/4}.$$

We want this to be upper bounded by $\varepsilon$, the final accuracy parameter, which means we can take

$$\varepsilon_{\mathrm{TV}} = c \frac{\varepsilon^4 H^{-8} K^{-4} d^{-2}}{\log^2(1 + {}^1\!/\!\varepsilon)},$$

where $c > 0$ is a universal constant. Looking at the definition of $\varepsilon_{\mathrm{TV}}$ and $T_{\max}$, we set

$$\lambda = c \frac{\varepsilon^4 H^{-8} K^{-4} d^{-3}}{\log^2(1 + {}^1\!/\!\varepsilon)}, \quad T_{\max} = \tilde{O}\left( \frac{H^{13} d^5 K^5}{\varepsilon^6} \right), \quad n = \tilde{O}\left( \frac{H^8 K^4 d^2 \log(|\Phi||\Upsilon|/\delta)}{\varepsilon^4} \right),$$

where we are ignoring logarithmic factors. This gives the final sample complexity of

$$\tilde{O}\left( \frac{H^{22} K^9 d^7 \log(|\Phi||\Upsilon|/\delta)}{\varepsilon^{10}} \right).$$

Finally, note that (1) is implied by the final accuracy guarantee, since we may choose $R$ to be the total variation distance between our model and the true transition dynamics at time $h$, which is clearly a sparse reward function.

**Analysis with the sampling oracle.**   With the sampling oracle, the argument is very similar. The main difference is in Lemma 8, which is the only place where we use the exact planner. Instead, we modify the proof of Lemma 8 to instead use Lemma 17 to obtain $\widehat{\Sigma}$ and $\rho_h^{\mathrm{pre}}$, and we do the Cauchy-Schwarz step using $\widehat{\Sigma}$. By Lemma 17 the first term is still $O(T\beta)$ and for the second term we pay an additive $O(\beta)$ to translate from $\widehat{\Sigma}$ to $\Sigma$ (since the spectral norm error is $O({}^\beta\!/\!d)$ and squared Euclidean norm of the term involving $\widehat{\mu}_{h-1}$ is at most $d$). This we have an additional $O(\alpha\beta)$ term in the sample-based analog of Lemma 8. However, as we use the bound $T \leq O(d \log(1 + {}^1\!/\!\beta)/\beta)$ in the remaining calculations, the new $O(\alpha\beta)$ term is only larger than the $O(\alpha d/T)$ term by a logarithmic factor. In particular, above we have a $\sqrt{d\beta}$ term in the bound for $\varepsilon_{\mathrm{escape}}$, but with the sampling oracle, we will additionally have a $O(\sqrt{T}\beta) = O(\sqrt{d\beta \log(1 + {}^1\!/\!\beta)})$ term in this bound. Ultimately, this only affects the final sample complexity bound in logarithmic factors. We adjust the failure probability accordingly, using $\delta/2$ probability for invocations of Lemma 17, and $\delta/2$ for the invocations of Corollary 4. As we invoke the planner polynomially many times, the total number of calls to the sampling oracle is polynomial in all parameters.

### B.1   Refined analysis for simplex representations.

Here we prove Theorem 3 by considering a different potential function argument and a different instantiation of the planning algorithm that directly attempts to visit each latent state. In particular, we instantiate Algorithm 1 with the planning routine presented in Algorithm 3. Note that this planner does not require the parameter $\beta$, but it does assume that $\widehat{\phi}(x, a) \in \Delta([d_{\mathrm{LV}}])$ for each $(x, a)$.

The analog of $\Sigma_{h,j}$ is the cumulative probability of hitting latent variables $z_h \in \mathcal{Z}_h$. Formally, we define

$$p_{h,j}(z) := \sum_{i=0}^{j-1} \mathbb{P}\left[ z_h = z \mid \rho_i, \mathcal{M} \right].$$

We make two remarks. First $p_{h,j}$ is not a distribution, rather it is the sum of $j$ probability distributions. Second, we use $p_{h,j}$ to measure the coverage at time $h$, since $z_h$ is the latent variable that generates $x_h$. This indexing is different from how we use $\Sigma_{h,j}$ to measure coverage at time $h + 1$ in the general case.

We now state the analog of Corollary 4.

**Corollary 10.** *For $j \geq 1, h \in \{0, \ldots, H - 1\}, \delta \in (0,1)$ and let $\rho_0, \ldots, \rho_{j-1}$ be any (possibly data-dependent) policies, with $p_{h,j}$ defined accordingly. Let $D_h$ be a dataset of $nj$ examples, where*

*for each $0 \leq i < j$, we collect $n$ triples $(x_h, a_h, x_{h+1})$ by rolling in with $\rho_i$ to $x_h$ and taking $a_h$ uniformly at random. Then, with probability at least $1 - \delta$ the output $(\widehat{\phi}_h, \widehat{\mu}_h)$ of $\mathrm{MLE}(D_h)$ satisfies*

$$\sum_{z \in \mathcal{Z}_h} p_{h,j}(z) \mathbb{E}\left[ \left\| \left\langle \widehat{\phi}(x_h, a_h), \widehat{\mu}_h(\cdot) \right\rangle - T_h(\cdot \mid x_h, a_h) \right\|_{\mathrm{TV}}^2 \mid z_h = z, a_h \sim \mathrm{unif}(\mathcal{A}) \right] \leq \frac{2\log(|\Phi||\Upsilon|/\delta)}{n}.$$

*Proof.* This is an immediate consequence of Theorem 18, using the definition of $p_{h,j}$. $\qquad\square$

We denote the LHS of the above lemma as $\mathrm{err}_h(p_{h,j})$. Now we define the known set $\mathcal{K}_h$ and the absorbing MDP. In the simplex features setting, the known set $\mathcal{K}_h$ is instead defined in terms of latent variables. Recall that we can augment every trajectory $\tau$ with the latent variables generated along the trajectory that is $\tau = (z_0, x_0, a_0, z_1, x_1, a_1, \ldots, z_{H-1}, x_{H-1}, a_{H-1})$. We therefore define $\mathcal{K}_{h,j} := \{z \in \mathcal{Z}_h : p_{h,j}(z) \geq \Delta\}$ where $\Delta$ is some parameter we will set towards the end of the proof. The absorbing MDP $\mathcal{M}_{\mathcal{K}}$ in iteration $j$ is defined to have transition operator that, for each $h$, transitions from $z_h$ to $x_{\mathrm{absorb}}$ if $z_h \notin \mathcal{K}_{h,j}$ and otherwise transitions as in $\mathcal{M}$. As in the more general analysis, $x_{\mathrm{absorb}}$ is an absorbing state with a single self-looping action $a_{\mathrm{absorb}}$ and we always consider $(x_{\mathrm{absorb}}, a_{\mathrm{absorb}})$ to be known.

We now state the analog of Lemma 5.

**Lemma 11.** *Let $\widehat{\phi}_{0:H-1}, \widehat{\mu}_{0:H-1}$ be an MDP model with simplex features and let $p_{0:H-1}$ be non-negative vectors. Assume that $\mathrm{err}_h(p_{h-1}) \leq \varepsilon_{\mathrm{TV}}$ for each $h$. Let $f : \mathcal{X} \times \mathcal{A} \to [0, 1]$ be any function such that $f(x_{\mathrm{absorb}}, a_{\mathrm{absorb}}) = 0$ and let $\pi$ be any policy. Then, for any $h$*

$$\mathbb{E}_\pi\left[f(x_h, a_h) \mid \mathcal{M}_{\mathcal{K}}\right] - HK\sqrt{\varepsilon_{\mathrm{TV}}/\Delta} \leq \widehat{\mathbb{E}}_\pi\left[f(x_h, a_h)\right] \leq \mathbb{E}_\pi\left[f(x_h, a_h) \mid \mathcal{M}_{\mathcal{K}}\right] + HK\sqrt{\varepsilon_{\mathrm{TV}}/\Delta}$$
$$+ \sum_{h'=1}^h \mathbb{P}\left[z_{h'} \notin \mathcal{K}_{h'} \mid \pi, \mathcal{M}_{\mathcal{K}}\right]$$

*Proof.* As in the proof of Lemma 5, we must control two terms for each $h' < h$:

$$\mathrm{Term}1_{h'} := \mathbb{E}\left[\int (\widehat{T}_{h'}(x_{h'+1} \mid x_{h'}, a_{h'}) - T_{h'}(x_{h'+1} \mid x_{h'}, a_{h'}))\widehat{V}_{h'+1}(x_{h'+1}) \mid \pi, \mathcal{M}_{\mathcal{K}}\right]$$
$$\mathrm{Term}2_{h'} := \mathbb{E}\left[\int (T_{h'}(x_{h'+1} \mid x_{h'}, a_{h'}) - T_{h',\mathcal{K}}(x_{h'+1} \mid x_{h'}, a_{h'}))\widehat{V}_{h'+1}(x_{h'+1}) \mid \pi, \mathcal{M}_{\mathcal{K}}\right].$$

For $\mathrm{Term}1_{h'}$, as we are in $\mathcal{M}_{\mathcal{K}}$ we can ignore the trajectories where $z_{h'} \notin \mathcal{K}_{h'}$. Thus considering $h' = 1$

$$\mathrm{Term}1_1 = \sum_{z \in \mathcal{K}_1} \mathbb{P}_\pi[z_1 = z \mid \mathcal{M}_{\mathcal{K}}] \cdot \mathbb{E}_\pi\left[\int (\widehat{T}_1(x_2 \mid x_1, a_1) - T_1(x_2 \mid x_1, a_1))\widehat{V}_2(x_2) \mid z_1 = z\right]$$
$$\leq K \sum_{z \in \mathcal{K}_1} \mathbb{P}_\pi[z_1 = z \mid \mathcal{M}_{\mathcal{K}}] \cdot \mathbb{E}\left[\left\|\widehat{T}_1(\cdot \mid x_1, a_1) - T_1(\cdot \mid x_1, a_1)\right\|_{\mathrm{TV}} \mid z_1 = z, a_1 \sim \mathrm{unif}(\mathcal{A})\right]$$
$$\leq K \sqrt{\sum_{z \in \mathcal{K}_1} \mathbb{P}_\pi[z_1 = z \mid \mathcal{M}_{\mathcal{K}}] \cdot \mathbb{E}\left[\left\|\widehat{T}_1(\cdot \mid x_1, a_1) - T_1(\cdot \mid x_1, a_1)\right\|_{\mathrm{TV}}^2 \mid z_1 = z, a_1 \sim \mathrm{unif}(\mathcal{A})\right]}$$
$$\leq K\sqrt{\mathrm{err}_1(p_1)/\Delta} \leq K\sqrt{\varepsilon_{\mathrm{TV}}/\Delta}.$$

Here we are using that the total variation term is non-negative, and that $\mathbb{P}_\pi[z_1 = z \mid \mathcal{M}_{\mathcal{K}}] \leq 1$, while $p_1(z) \geq \Delta$ by the fact that $z \in \mathcal{K}_1$. This argument applies for all $h'$ and yields the $HK\sqrt{\varepsilon_{\mathrm{TV}}/\Delta}$ term on both sides of the statement. For $\mathrm{Term}2_{h'}$, we clearly have

$$\mathrm{Term}2_{h'} \leq \mathbb{P}\left[z_{h'+1} \notin \mathcal{K}_{h'+1} \mid \pi, \mathcal{M}_{\mathcal{K}}\right].$$

As in the proof of Lemma 5, $\mathrm{Term}2_{h'} \geq 0$ which yields the lower bound. $\qquad\square$

Next we argue that if the exploratory policy $\rho_j$ that we find has large escaping probability then we will add some latent variable to the known set in the next iteration.

**Lemma 12.** *Consider iteration $j$ of* FLAMBE *and assume that* $\mathrm{err}_h(p_{h,j}) \leq \varepsilon_{\mathrm{TV}}$ *for each $h$. Define* $R_h(x,a) := \sum_{z \notin \mathcal{K}_{h,j}} \phi_h^\star(x,a)[z]$. *Then*

$$\max_h \mathbb{P}\left[z_h \notin \mathcal{K}_{h,j} \mid \rho_j\right] \geq \frac{1}{H} \max_h \left\{ \widehat{\mathbb{E}}_{\rho_j}[R_h(x_h, a_h)] - HK\sqrt{\varepsilon_{\mathrm{TV}}/\Delta} \right\}.$$

*In particular, if there exists some $h$ such that* $\widehat{\mathbb{E}}_{\rho_j}[R_h(x_h, a_h)] \geq HK\sqrt{\varepsilon_{\mathrm{TV}}/\Delta} + Hd_{\mathrm{LV}}\Delta$, *then there exists some $h', z \notin \mathcal{K}_{h'}$ such that* $\mathbb{P}[z_{h'} = z \mid \rho_j] \geq \Delta$.

*Proof.* Observe that by the definition of $R_h$, we have

$$\widehat{\mathbb{E}}_{\rho_j}\left[R_h(x_h, a_h)\right] \leq \mathbb{E}_{\rho_j}[R_h(x_h, a_h) \mid \mathcal{M}_\mathcal{K}] + HK\sqrt{\varepsilon_{\mathrm{TV}}/\Delta} + \sum_{h'=1}^{h} \mathbb{P}\left[z_{h'} \notin \mathcal{K}_{h',j} \mid \rho_j, \mathcal{M}_\mathcal{K}\right]$$

$$= HK\sqrt{\varepsilon_{\mathrm{TV}}/\Delta} + \sum_{h'=1}^{h+1} \mathbb{P}\left[z_{h'} \notin \mathcal{K}_{h',j} \mid \rho_j, \mathcal{M}_\mathcal{K}\right].$$

Both statements now follow from the pigeonhole principle. $\qquad\square$

Next we prove the analog of Lemma 9. For this, we compute $\rho_h^{\mathrm{pre}}$ using the planning routine in Algorithm 3, with the planning guarantee in Lemma 15.

**Lemma 13.** *Assume that for each round $j \in [J_{\max}]$ and for all $h$, we have* $\mathrm{err}_h(p_{h,j}) \leq \varepsilon_{\mathrm{TV}}$ *and set $J_{\max} = Hd_{\mathrm{LV}} + 1$. Then the final MDP model $\widehat{\mathcal{M}}$ satisfies the following guarantee: For any sparse reward function $R : \mathcal{X} \times \mathcal{A} \to [0,1]$ and any policy $\pi$, we have*

$$\left| V(\pi; R, \widehat{\mathcal{M}}) - V(\pi; R, \mathcal{M}) \right| \leq HK\sqrt{\varepsilon_{\mathrm{TV}}/\Delta} + H\varepsilon_{\mathrm{escape}},$$

*where* $\varepsilon_{\mathrm{escape}} := H^2 K d_{\mathrm{LV}}^2 \Delta + \left(H^2 K^2 d_{\mathrm{LV}} + HK d_{\mathrm{LV}}\right)\sqrt{\varepsilon_{\mathrm{TV}}/\Delta}$.

*Proof.* First observe that by Lemma 12, in every iteration where $\rho_j$ satisfies $\max_h \widehat{\mathbb{E}}_{\rho_j}[R_h(x_h, a_h)] \geq HK\sqrt{\varepsilon_{\mathrm{TV}}/\Delta} + Hd_{\mathrm{LV}}\Delta$, we add some latent variable at some time point to the known set. This means that this can only happen for at most $Hd_{\mathrm{LV}}$ iterations, and so, by the setting of $J_{\max}$, there must be some iteration $j$ in which $\max_h \widehat{\mathbb{E}}_{\rho_j}[R_h(x_h, a_h)] \leq HK\sqrt{\varepsilon_{\mathrm{TV}}/\Delta} + Hd_{\mathrm{LV}}\Delta$. In this iteration $j$, we have

$$\mathbb{P}\left[z_{h+1} \notin \mathcal{K}_{h+1,j} \mid \pi, \mathcal{M}_\mathcal{K}^{(j)}\right] \leq \mathbb{P}\left[z_{h+1} \notin \mathcal{K}_{h+1,j} \mid \pi, \widehat{\mathcal{M}}^{(j)}\right] + HK\sqrt{\varepsilon_{\mathrm{TV}}/\Delta}$$

$$= \sum_{z \in \widehat{\mathcal{Z}}_h} \widehat{\mathbb{E}}_\pi\left[\widehat{\phi}_{h-1}(x_{h-1}, a_{h-1})[z]\right] \cdot \mathbb{P}\left[\bar{\mathcal{K}}_{h+1,j} \mid \pi, \widehat{\mathcal{M}}^{(j)}, \widehat{z}_h = i\right] + HK\sqrt{\varepsilon_{\mathrm{TV}}/\Delta}$$

$$\leq K \sum_{z \in \widehat{\mathcal{Z}}_h} \widehat{\mathbb{E}}_\pi\left[\widehat{\phi}_{h-1}(x_{h-1}, a_{h-1})[z]\right] \cdot \mathbb{P}\left[\bar{\mathcal{K}}_{h+1,j} \mid \widehat{\mathcal{M}}^{(j)}, \widehat{z}_h = z, a_h \sim \mathrm{unif}(\mathcal{A})\right] + HK\sqrt{\varepsilon_{\mathrm{TV}}/\Delta}$$

$$\leq K d_{\mathrm{LV}} \sum_{z \in \widehat{\mathcal{Z}}_h} \widehat{\mathbb{E}}_{\rho_h^{\mathrm{pre}}}\left[\widehat{\phi}_{h-1}(x_{h-1}, a_{h-1})[z]\right] \cdot \mathbb{P}\left[\bar{\mathcal{K}}_{h+1,j} \mid \widehat{\mathcal{M}}^{(j)}, \widehat{z}_h = z, a_h \sim \mathrm{unif}(\mathcal{A})\right] + HK\sqrt{\varepsilon_{\mathrm{TV}}/\Delta}$$

$$\leq HK d_{\mathrm{LV}} \widehat{\mathbb{P}}_{\rho_j}\left[\bar{\mathcal{K}}_{h+1,j}\right] + HK\sqrt{\varepsilon_{\mathrm{TV}}/\Delta} \leq HK d_{\mathrm{LV}} \widehat{\mathbb{E}}_{\rho_j}[R_h(x_h, a_h)] + HK\sqrt{\varepsilon_{\mathrm{TV}}/\Delta}$$

$$\leq HK d_{\mathrm{LV}}\left(HK\sqrt{\varepsilon_{\mathrm{TV}}/\Delta} + Hd_{\mathrm{LV}}\Delta\right) + HK\sqrt{\varepsilon_{\mathrm{TV}}/\Delta} =: \varepsilon_{\mathrm{escape}}$$

Here the first inequality is Lemma 11, while the first equality re-writes the expectation in terms of the latent variable $z_h$. In the second inequality we translate to taking $a_h$ uniformly via importance weighting, while in the third, we apply Lemma 15, which lets us translate to $\rho_h^{\mathrm{pre}}$. Finally, we use that $\rho_j$ uses $\rho_h^{\mathrm{pre}}$ with probability $1/H$ and the definition of $R_h$. The result now follows from Lemma 11, along with the analog of (3). As in the general case, this bound applies for all iterations after the first one where the escaping probability for $\rho_j$ is small. $\qquad\square$

**Final steps.** The final steps with simplex features are much more straightforward than in the general case. First we choose $\Delta$ to balance the two terms in $\varepsilon_{\text{escape}}$. We set $\Delta = (K/d_{\text{LV}})^{2/3}\varepsilon_{\text{TV}}^{1/3}$ which yields

$$\varepsilon_{\text{escape}} \leq 3H^2 K d_{\text{LV}}\left(K^{2/3}(d_{\text{LV}}\varepsilon_{\text{TV}})^{1/3}\right),$$

where we are also using the fact that $\varepsilon_{\text{TV}} \leq 1$. Via Lemma 13, after $J_{\max} = H d_{\text{LV}} + 1$ iterations, we are guaranteed that

$$\max_{\pi,R}\left|V(\pi; R, \widehat{\mathcal{M}}) - V(\pi; R, \mathcal{M})\right| \leq HK\sqrt{\varepsilon_{\text{TV}}/\Delta} + H\varepsilon_{\text{escape}} \leq O\left(H^3 K^{5/3} d_{\text{LV}}^{4/3}\varepsilon_{\text{TV}}^{1/3}\right).$$

For this to be at most $\varepsilon$ we should set $\varepsilon_{\text{TV}} \leq O\left(\varepsilon^3 H^{-9} K^{-5} d_{\text{LV}}^{-4}\right)$. Applying Corollary 10 and taking a union over all $J_{\max}$ rounds, we want to set

$$n = \frac{2\log(J_{\max}|\Phi||\Upsilon|/\delta)}{\varepsilon_{\text{TV}}} = \tilde{O}\left(\frac{H^9 K^5 d_{\text{LV}}^4 \log(|\Phi||\Upsilon|/\delta)}{\varepsilon^3}\right).$$

The total sample complexity is $nHJ_{\max} = \tilde{O}\left(\frac{H^{11} K^5 d_{\text{LV}}^5 \log(|\Phi||\Upsilon|/\delta)}{\varepsilon^3}\right)$. As in the general setting, the value function guarantee implies (1), which yields the result.

**Analysis with a sampling oracle.** With a sampling oracle, the only difference is in the proof of Lemma 13. Here, we can only apply the second statement of Lemma 15, which yields an additive $O(Kd_{\text{LV}}\varepsilon_{\text{opt}})$ term. By taking $\varepsilon_{\text{opt}} = (H/d_{\text{LV}})\sqrt{\varepsilon_{\text{TV}}/\Delta}$, this additional term can be absorbed into the other additive term at the expense of a constant. Thus we obtain the same guarantee, up to constants, with polynomially many calls to the sampling oracle.

## C  Planning Algorithms

In this section, we present exploratory planning algorithms for low rank models, assuming that the dynamics are known. Formally, we consider an $H$ step low rank MDP $\widetilde{\mathcal{M}}$ with deterministic start state $x_0$, fixed action $a_0$, and transition matrices $T_0, \ldots, T_{H-1}$. Each transition operator $T_h$ factorizes as $T_h(x_{h+1} \mid x_h, a_h) = \langle \phi_h(x_h, a_h), \mu_h(x_{h+1})\rangle$ and we assume $\phi_{0:H-1}, \mu_{0:H-1}$ are *known*. To compartmentalize the results, we focus on exploratory planning at time $H$, but we will invoke these subroutines with MDP models that have horizon $h \leq H$. This simply requires rebinding variables.

We present two types of results. One style assumes that all expectations are computed exactly. As we are focusing purely on planning with known dynamics and rewards, this imposes a computational burden, but not a statistical one, while leading to a more transparent proof. To address the computational burden, we also consider algorithms that approximate all expectations with samples. For this, we assume that we can obtain sample transitions from the MDP model $\widetilde{\mathcal{M}}$ in a computationally efficient manner. Formally, the *sampling oracle* allows us to sample $x' \sim T_h(\cdot \mid x, a)$ for any $x, a$.

### C.1  Planning with a sampling oracle

For the computational style of result, it will be helpful to first show how to optimize a given reward function whenever the model admits a sampling oracle. As notation, we always consider an explicitly specified non-stationary reward function $R : \mathcal{X} \times \mathcal{A} \times \{0, \ldots, H-1\} \to [0, 1]$. Then, we define

$$V(\pi, R) = \mathbb{E}\left[\sum_{h=0}^{H-1} R(x_h, a_h, h) \mid \pi, \widetilde{\mathcal{M}}\right].$$

The next lemma is a simple application of the result of Jin et al. (2020b).

**Lemma 14.** *Suppose that the reward function $R : \mathcal{X} \times \mathcal{A} \times \{0, \ldots, H-1\} \to [0, 1]$ is explicitly given and that $T_{0:H-1}$ is a known low rank MDP that enables efficient sampling. Then for any $\epsilon > 0$ there is an algorithm for finding a policy $\widehat{\pi}$ such that with probability at least $1 - \delta$, $V(\widehat{\pi}, R) \geq \max_\pi V(\pi, R) - \epsilon$ in polynomial time with $\text{poly}(d, H, 1/\epsilon, \log(1/\delta))$ calls to the sampling routine.*

---

**Algorithm 3** Exploratory planner for simplex representations

---

1: **Input:** MDP $\widetilde{\mathcal{M}} = (\phi_{0:H-1}, \mu_{0:H-1})$ with $\phi_h(x_h, a_h) \in \Delta([d_{\mathrm{LV}}])$, $\mu_h[z] \in \Delta(\mathcal{X})$.
2: **for** $z = 1, \ldots, d_{\mathrm{LV}}$ **do**
3:      Compute $\pi_z = \mathrm{argmax}_\pi \mathbb{E}[\phi_{H-1}(x_{H-1}, a_{H-1})[z] \mid \widetilde{\mathcal{M}}, \pi]$
4: **end for**
5: Output policy mixture $\rho := \mathrm{unif}(\{\pi_z\}_{z=1}^{d_{\mathrm{LV}}})$

---

*Proof.* As we have sampling access to the MDP, we can execute the LSVI-UCB algorithm of Jin et al. (2020b). For any $n$, if we execute the algorithm for $n$ episodes, it produces $n$ policies $\pi_1, \ldots, \pi_n$ and guarantees

$$\max_\pi V(\pi, R) - \frac{1}{n} \sum_{i=1}^n V(\pi_i, R) \leq c \sqrt{\frac{d^3 H^3 \log(n d H / \delta)}{n}}$$

with probability at least $1 - \delta$ where $c > 0$ is a universal constant. We are assured that one of the policies $\pi_1, \ldots, \pi_n$ is at most $\epsilon/2$-suboptimal by taking $n = O\left(d^3 H^3 \log(dH/(\epsilon\delta))/\epsilon^2\right)$.

We find this policy via a simple policy evaluation step. For each policy $\pi_i$, we collect $O(H^2 \log(n/\delta)/\epsilon^2)$ roll-outs using the generative model, where we take actions according to $\pi_i$. Via a union bound, this guarantees that for each $i$ we have $\widehat{V}_i$ such that with probability at least $1 - \delta$

$$\max_i \left| \widehat{V}_i - V(\pi_i, R) \right| \leq \epsilon/4.$$

Therefore, if we take $\widehat{i} = \mathrm{argmax}_{i \in [n]} \widehat{V}_i$ we are assured that $V(\pi_{\widehat{i}}, R) \geq \max_\pi V(\pi, R) - \epsilon$ with probability at least $1 - 2\delta$. The total number of samples required from the model are

$$nH \left( 1 + \frac{H^2 \log(n/\delta)}{\epsilon^2} \right) = \tilde{O}\left( \frac{d^3 H^6 \log(1/\delta)}{\epsilon^4} \right). \qquad \square$$

### C.2 Planning with simplex features

We first consider a simpler planning algorithm that is adapted to the simplex features representation. The pseudocode is displayed in Algorithm 3. The planner computes a mixture policy $\rho$, where component $\pi_i$ of the mixture focuses on activating coordinate $i$ of the feature map $\phi_{H-1}(x_{H-1}, a_{H-1})$. Each mixture component can be computed in a straightforward manner using a dynamic programming approach, such as LSVI. The basic guarantee for this algorithm is the following lemma.

**Lemma 15** (Guarantee for Algorithm 3). *If $\widetilde{\mathcal{M}}$ is an $H$-step low rank MDP with simplex features of dimension $d_{\mathrm{LV}}$, then the output of Algorithm 3, $\rho$, satisfies*

$$\forall \pi, z \in [d_{\mathrm{LV}}] : \mathbb{E}\left[ \phi_{H-1}(x_{H-1}, a_{H-1})[z] \mid \widetilde{\mathcal{M}}, \pi \right] \leq d_{\mathrm{LV}} \mathbb{E}\left[ \phi_{H-1}(x_{H-1}, a_{H-1})[z] \mid \widetilde{\mathcal{M}}, \rho \right].$$

*Given a sampling oracle for $\widetilde{\mathcal{M}}$, the algorithm runs in polynomial time with* $\mathrm{poly}(d_{\mathrm{LV}}, H, 1/\varepsilon_{\mathrm{opt}}, \log(1/\delta))$ *calls to* SAMP, *and with probability at least $1 - \delta$, $\rho$ satisfies*

$$\forall \pi, z \in [d_{\mathrm{LV}}], \mathbb{E}\left[ \phi_{H-1}(x_{H-1}, a_{H-1})[z] \mid \widetilde{\mathcal{M}}, \pi \right] \leq d_{\mathrm{LV}} \mathbb{E}\left[ \phi_{H-1}(x_{H-1}, a_{H-1})[z] \mid \widetilde{\mathcal{M}}, \rho \right] + \varepsilon_{\mathrm{opt}}.$$

*Proof.* The first result follows immediately from the non-negativity of $\phi_{H-1}(x_{H-1}, a_{H-1})[i]$, the optimality property of $\pi_i$ and the definition of $\rho$.

For the second result, by Lemma 14 we can optimize any explicitly specified reward function using a polynomial number of samples. If we call this sampling-based planner for each of the $d$ reward functions, with high probability (via a union bound) the policies $\widehat{\pi}_i$ are near-optimal for their corresponding reward functions. By appropriately re-scaling the accuracy parameter in Lemma 14 we obtain the desired guarantee. $\qquad \square$

## C.3 Elliptical planner

The next planning algorithm applies to general low rank MDP, and it is more sophisticated. It proceeds in iterations, where in iteration $t$ we maintain a covariance matrix $\Sigma_{t-1}$ and, in (2), we search for a policy that maximizes quadratic forms with the inverse covariance $\Sigma_{t-1}^{-1}$. With a sampling oracle this optimization can be done via a call to Lemma 14. If this maximizing policy $\pi_t$ cannot achieve large quadratic forms against $\Sigma_{t-1}^{-1}$, then we halt and output the mixture of all previous policies. Otherwise, we mix $\pi_t$ into our candidate solution, update the covariance matrix accordingly, and advance to the next iteration. The performance guarantee for this algorithm is as follows.

**Lemma 16** (Guarantee for Algorithm 2). *If $\widetilde{\mathcal{M}}$ is an $H$-step low rank MDP with embedding dimension $d$ then for any $\beta > 0$, Algorithm 2 terminates after at most $T + 1$ iterations where $T \leq 4d \log(1 + 4/\beta)/\beta$. Upon termination, $\rho$ guarantees*

$$\forall \pi : \mathbb{E}\left[\phi_{H-1}(x_{H-1}, a_{H-1})^\top \left(\Sigma_\rho + I/T\right)^{-1} \phi_{H-1}(x_{H-1}, a_{H-1}) \mid \widetilde{\mathcal{M}}, \pi\right] \leq T\beta.$$

*where $\Sigma_\rho = \frac{1}{T}\sum_{t=1}^T \Sigma_{\pi_t}$.*

*Proof.* The performance guarantee is immediate from the termination condition, using the fact that $\Sigma_T = T \cdot (\Sigma_\rho + I/T)$.

For the iteration complexity bound, we condense the notation and omit the dependence on $H - 1$, $x_{H-1}, a_{H-1}$ in all terms. We have

$$\beta T \leq \sum_{t=1}^T \mathbb{E}\left[\phi^\top \Sigma_{t-1}^{-1} \phi \mid \widetilde{\mathcal{M}}, \pi_t\right] = \sum_{t=1}^T \mathrm{tr}(\Sigma_{\pi_t}\Sigma_{t-1}^{-1}) \leq 2d\log(1 + T/d),$$

where the first inequality is based on the fact that we did not terminate at each iteration $t \in [T]$ and the last inequality follows from a standard elliptical potential argument (e.g., Lemma 11 in Dani et al. (2008); see Lemma 23 for a precise statement and proof). This gives an upper bound on $T$ that implies the one in the lemma statement, via Corollary 24. $\qquad\square$

With the sampling oracle, we modify the algorithm slightly and obtain a qualitatively similar guarantee. The modifications are discussed in the proof.

**Lemma 17.** *The sample-based version of Algorithm 2 has the following guarantee. Assume $\widetilde{\mathcal{M}}$ is an $H$-step low rank MDP with embedding dimension $d$ and fix $\beta > 0$, $\delta \in (0, 1)$. Then the algorithm terminates after at most $T + 1$ iterations, where $T \leq O(d \log(1 + 1/\beta)/\beta)$. Upon termination, it ouputs a matrix $\widehat{\Sigma}$ and a policy $\rho$ such that with probability at least $1 - \delta$:*

$$\forall \pi : \mathbb{E}\left[\phi_{H-1}(x_{H-1}, a_{H-1})^\top \left(\widehat{\Sigma} + I/T\right)^{-1} \phi_{H-1}(x_{H-1}, a_{H-1}) \mid \widetilde{\mathcal{M}}, \pi\right] \leq O(T\beta),$$

$$\left\|\widehat{\Sigma} - \left(\mathbb{E}\left[\phi_{H-1}(x_{H-1}, a_{H-1})\phi_{H-1}(x_{H-1}, a_{H-1})^\top \mid \rho, \widetilde{\mathcal{M}}\right] + I/T\right)\right\|_{\mathrm{op}} \leq O(\beta/d).$$

*The algorithm runs in polynomial time with $\mathrm{poly}(d, H, 1/\beta, \log(1/\delta))$ calls to the sampling oracle.*

*Proof.* The algorithm is modified as follows. We replace all covariances with empirical approximations, obtained by calls to the sampling subroutine. We call the empirical versions $\widehat{\Sigma}_t, \widehat{\Sigma}_{\pi_t}$, etc. Then, the policy optimization step (2) is performed via an application of Lemma 14 and so we find an $\varepsilon_{\mathrm{opt}}$-suboptimal policy $\pi_t$ for the reward function induced by $\widehat{\Sigma}_{t-1}$. Then we use the sampling subroutine to estimate the value of this policy, which we denote $\widehat{V}_t(\pi_t)$. As before, we terminate if $\widehat{V}_t(\pi_t) \leq \beta$. If we terminate in round $t$, we output $\rho = \mathrm{unif}(\{\pi_i\}_{i=1}^{t-1})$ and we also output $\widehat{\Sigma} = \frac{1}{t-1}\sum_{i=1}^{t-1} \widehat{\Sigma}_{\pi_i}$. As notation, we use $V_t(\pi)$ to denote the value for policy $\pi$ on the reward function used in iteration $t$, which is induced by $\widehat{\Sigma}_{t-1}$.

With $\mathrm{poly}(d, H, T, 1/\varepsilon_{\mathrm{opt}}, \log(1/\delta))$ calls to the sampling subroutine and assuming the total number of iterations of the algorithm $T$ is polynomial, we can verify that with probability $1 - \delta$

$$\max_{t \in [T]} \max \left\{ d \cdot \left\|\widehat{\Sigma}_{\pi_t} - \Sigma_{\pi_t}\right\|_{\mathrm{op}}, \left|\widehat{V}_t(\pi_t) - V_t(\pi_t)\right|, \max_\pi V_t(\pi) - V_t(\pi_t) \right\} \leq \varepsilon_{\mathrm{opt}}.$$

The first two bounds follow from standard concentration of measure arguments. The final one is based on an application of Lemma 14.

Now, if we terminate in iteration $t$, we know that $\widehat{V}_t(\pi_t) \leq \beta$. This implies

$$\max_\pi V_t(\pi) \leq V_t(\pi_t) + \varepsilon_{\mathrm{opt}} \leq \widehat{V}_t(\pi_t) + 2\varepsilon_{\mathrm{opt}} \leq \beta + 2\varepsilon_{\mathrm{opt}}.$$

As we are interested in the reward function induced by $\widehat{\Sigma}_{t-1}$, this verifies the quality guarantee, provided $\varepsilon_{\mathrm{opt}} = O(\beta)$.

Finally, we turn to the iteration complexity. Similarly to above, we have

$$
\begin{aligned}
T\left(\beta - 2\varepsilon_{\mathrm{opt}}\right) \leq \sum_{t=1}^{T} \widehat{V}_t(\pi_t) - 2\varepsilon_{\mathrm{opt}} &\leq \sum_{t=1}^{T} V_t(\pi_t) - \varepsilon_{\mathrm{opt}} \\
&= \sum_{t=1}^{T} \mathbb{E}\left[\phi^\top \widehat{\Sigma}_{t-1}^{-1} \phi \mid \widetilde{\mathcal{M}}, \pi_t\right] - \varepsilon_{\mathrm{opt}} = \sum_{t=1}^{T} \mathrm{tr}(\Sigma_{\pi_t} \widehat{\Sigma}_{t-1}^{-1}) - \varepsilon_{\mathrm{opt}} \\
&\leq \sum_{t=1}^{T} \mathrm{tr}(\widehat{\Sigma}_{\pi_t} \widehat{\Sigma}_{t-1}^{-1}) \leq 2d \log(1 + {}^T\!/_d).
\end{aligned}
$$

In other words, if we set $\varepsilon_{\mathrm{opt}} = O(\beta)$ then both the iteration complexity and the performance guarantee are unchanged. The accuracy guarantee for the covariance matrix $\widehat{\Sigma}_{t-1}$ is straightforward, since each $\widehat{\Sigma}_{\pi_t}$ is $\varepsilon_{\mathrm{opt}}$ accurate and $\widehat{\Sigma}$ is the average of such matrices. $\qquad\square$

## D  Maximum Likelihood Estimation

In this section we adapt classical results for maximum likelihood estimation in general parametric models. We consider a sequential conditional probability estimation setting with an instance space $\mathcal{X}$ and target space $\mathcal{Y}$ and with a conditional density $p(y \mid x) = f^\star(x, y)$. We are given a function class $\mathcal{F} : (\mathcal{X} \times \mathcal{Y}) \to \mathbb{R}$ with which to model the condition distribution $f^\star$, and we assume that $f^\star \in \mathcal{F}$, so that the problem is well-specified or realizable. We are given a dataset $D := \{(x_i, y_i)\}_{i=1}^{n}$, where $x_i \sim \mathcal{D}_i = \mathcal{D}_i(x_{1:i-1}, y_{1:i-1})$ and $y_i \sim p(\cdot \mid x_i)$. Note that $\mathcal{D}_i$ depends on the previous examples, so this is a martingale process. We optimize the maximum likelihood objective

$$\widehat{f} := \operatorname*{argmax}_{f \in \mathcal{F}} \sum_{i=1}^{n} \log f(x_i, y_i). \tag{6}$$

The iid version of the following result is classical (c.f., Van de Geer, 2000, Chapter 7), but under-utilized in machine learning and reinforcement learning in particular. Our adaptation is inspired by Zhang (2006).

**Theorem 18.** *Fix $\delta \in (0, 1)$, assume $|\mathcal{F}| < \infty$ and $f^\star \in \mathcal{F}$. Then with probability at least $1 - \delta$*

$$\sum_{i=1}^{n} \mathbb{E}_{x \sim \mathcal{D}_i} \left\| \widehat{f}(x, \cdot) - f^\star(x, \cdot) \right\|_{\mathrm{TV}}^2 \leq 2 \log(|\mathcal{F}|/\delta).$$

**Remark 19.** *Given a class of discriminators $\mathcal{G} : (\mathcal{X}, \mathcal{Y}) \mapsto [-1, 1]$, an alternative is to consider the following (conditional) "generative adversarial" objective:*

$$\widehat{f} = \operatorname*{argmin}_{f \in \mathcal{F}} \max_{g \in \mathcal{G}} \frac{1}{n} \sum_{i=1}^{n} \left( g(x_i, y_i) - \mathbb{E}[g(x_i, y) \mid y \sim f(x, \cdot)] \right).$$

*This is the natural objective associated with the distance function induced by $\mathcal{G}$ (Arora et al., 2017), and is also related to other GAN-style approaches. Owing to the realizability assumption, $f^\star$ will always have low objective value, scaling with the complexity of $\mathcal{G}$. Additionally, if $\mathcal{G}$ is expressive enough, one can establish a guarantee similar to Theorem 18, which can then be used in the analysis of FLAMBE. Formally, a sufficient condition is that $\mathcal{G}$ contains the indicators of the Scheffe sets for all pairs $f, f' \in \mathcal{F}$, in which case the total variation guarantee can be obtained by standard uniform convergence arguments. See Devroye and Lugosi (2012); Sun et al. (2019) for more details.*

**Remark 20.** *We also remark that the proof of [Theorem 18](#) actually establishes convergence in the squared Hellinger distance. We obtain the total variation guarantee simply by observing that the squared Hellinger distance dominates the squared total variation distance.*

We prove [Theorem 18](#) in this section. We begin with a decoupling inequality. Let $D$ denote the dataset and let $D'$ denote a *tangent sequence* $\{(x_i', y_i')\}_{i=1}^n$ where $x_i' \sim \mathcal{D}_i(x_{1:i-1}, y_{1:i-1})$ and $y_i' \sim p(\cdot \mid x_i')$. Note here that $x_i'$ depends on the original sequence, and so the tangent sequence is independent conditional on $D$.

**Lemma 21.** *Let $D$ be a dataset of $n$ examples, and let $D'$ be a tangent sequence. Let $L(f, D) = \sum_{i=1}^n \ell(f, (x_i, y_i))$ be any function that decomposes additively across examples where $\ell$ is any function, and let $\widehat{f}(D)$ be any estimator taking as input random variable $D$ and with range $\mathcal{F}$. Then*

$$\mathbb{E}_D \left[ \exp \left( L(\widehat{f}(D), D) - \log \mathbb{E}_{D'} \exp(L(\widehat{f}(D), D')) - \log |\mathcal{F}| \right) \right] \leq 1$$

Observe that in the second term, the "loss function" takes as input $D'$, but the estimator takes as input $D$. As such, the above inequality decouples the estimator from the loss.

*Proof.* Let $\pi$ be the uniform distribution over $\mathcal{F}$ and let $g : \mathcal{F} \to \mathbb{R}$ be any function. Define $\mu(f) := \frac{\exp(g(f))}{\sum_f \exp(g(f))}$, which is clearly a probability distribution. Now consider any other probability distribution $\widehat{\pi}$ over $\mathcal{F}$:

$$0 \leq \mathrm{KL}(\widehat{\pi} \| \mu) = \sum_f \widehat{\pi}(f) \log(\widehat{\pi}(f)) + \sum_f \widehat{\pi}(f) \log \left( \sum_{f'} \exp(g(f')) \right) - \sum_f \widehat{\pi}(f) g(f)$$

$$= \mathrm{KL}(\widehat{\pi} \| \pi) - \sum_f \widehat{\pi}(f) g(f) + \log \mathbb{E}_{f \sim \pi} \exp(g(f))$$

$$\leq \log |\mathcal{F}| - \sum_f \widehat{\pi}(f) g(f) + \log \mathbb{E}_{f \sim \pi} \exp(g(f)).$$

Re-arranging, it holds that

$$\sum_f \widehat{\pi}(f) g(f) - \log |\mathcal{F}| \leq \log \mathbb{E}_{f \sim \pi} \exp(g(f)).$$

We instantiate this bound with $\widehat{\pi} = \mathbf{1}\{\widehat{f}(D)\}$ and $g(f) = L(f, D) - \log \mathbb{E}_{D'} \exp(L(f, D'))$ to obtain, for any $D$

$$L(\widehat{f}(D), D) - \log \mathbb{E}_{D'} \exp(L(\widehat{f}(D), D')) - \log |\mathcal{F}| \leq \log \mathbb{E}_{f \sim \pi} \frac{\exp(L(f, D))}{\mathbb{E}_{D'} \exp(L(f, D'))}.$$

Exponentiating both sides and then taking expectation over $D$, we obtain

$$\mathbb{E}_D \left[ \exp(L(\widehat{f}(D), D) - \log \mathbb{E}_{D'} \exp(L(\widehat{f}(D), D')) - \log |\mathcal{F}|) \right]$$

$$\leq \mathbb{E}_{f \sim \pi} \mathbb{E}_D \frac{\exp(L(f, D))}{\mathbb{E}_{D'}[\exp(L(f, D')) \mid D]} = 1.$$

The last equality follows since, conditional on $D$, the tangent sequence $D'$ is independent. Therefore,

$$\mathbb{E}_{D'} \left[ \exp(L(f, D')) \mid D \right] = \prod_{i=1}^n \mathbb{E}_{(x_i', y_i') \sim \mathcal{D}_i} \left[ \exp(\ell(f, (x_i', y_i'))) \right],$$

which allows us to peel off terms starting from $n$ down to $1$ and cancel them with those in the numerator. $\qquad\square$

The next lemma translates from TV-distance to a loss function that is closely related to the KL divergence.

**Lemma 22.** *For any two conditional probability densities $f_1, f_2$ and any distribution $\mathcal{D} \in \Delta(\mathcal{X})$ we have*

$$\mathbb{E}_{x \sim \mathcal{D}} \|f_1(x, \cdot) - f_2(x, \cdot)\|_{\mathrm{TV}}^2 \leq -2 \log \mathbb{E}_{x \sim \mathcal{D}, y \sim f_2(\cdot|x)} \exp\left(-\frac{1}{2} \log(f_2(x, y)/f_1(x, y))\right)$$

*Proof.* Let us begin by relating the total variation distance, which appears on the left hand side, to the (squared) Hellinger distance, which for densities $p, q$ over a domain $\mathcal{Z}$ is defined as

$$\mathrm{H}^2(q\|p) := \int \left(\sqrt{p(z)} - \sqrt{q(z)}\right)^2 dz.$$

Lemma 2.3 in Tsybakov (2008) asserts that

$$\|p(\cdot) - q(\cdot)\|_{\mathrm{TV}}^2 \leq \mathrm{H}^2(q\|p) \cdot \left(1 - \frac{\mathrm{H}^2(q\|p)}{4}\right) \leq \mathrm{H}^2(q\|p),$$

where the final inequality uses non-negativity of the Hellinger distance. Next, note that we can also write

$$\mathrm{H}^2(q\|p) = \int p(z) + q(z) - 2\sqrt{p(z)q(z)} dz = 2 \cdot \mathbb{E}_{z \sim q}\left[1 - \sqrt{p(z)/q(z)}\right]$$

$$\leq -2 \log \mathbb{E}_{z \sim q} \sqrt{p(z)/q(z)} = -2 \log \mathbb{E}_{z \sim q} \exp\left(-\frac{1}{2} \log(q(z)/p(z))\right).$$

Here the inequality follows from the fact that $1 - x \leq -\log(x)$. The result follows by applying this argument to $\mathbb{E}_{x \sim \mathcal{D}} \|f_1(x, \cdot) - f_2(x, \cdot)\|_{\mathrm{TV}}^2$. $\qquad\square$

*Proof of Theorem 18.* First note that Lemma 21 can be combined with the Chernoff method to obtain an exponential tail bound: with probability $1 - \delta$ we have

$$-\log \mathbb{E}_{D'} \exp(L(\widehat{f}(D), D')) \leq -L(\widehat{f}(D), D) + \log|\mathcal{F}| + \log(1/\delta).$$

Now we set $L(f, D) = \sum_{i=1}^n -1/2 \cdot \log(f^\star(x_i, y_i)/f(x_i, y_i))$ where $D$ is a dataset $\{(x_i, y_i)\}_{i=1}^n$ (and $D' = \{(x_i', y_i')\}_{i=1}^n$ is a tangent sequence). With this choice, the right hand side is

$$\sum_{i=1}^n \frac{1}{2} \log(f^\star(x_i, y_i)/\widehat{f}(x_i, y_i)) + \log|\mathcal{F}| + \log(1/\delta) \leq \log|\mathcal{F}| + \log(1/\delta),$$

since $\widehat{f}$ is the empirical maximum likelihood estimator and we are in the well-specified setting. On the other hand, the left hand side is

$$-\log \mathbb{E}_{D'}\left[\exp\left(\sum_{i=1}^n -1/2 \log\left(\frac{f^\star(x_i', y_i')}{\widehat{f}(x_i', y_i')}\right)\right) \mid D\right] = -\sum_{i=1}^n \log \mathbb{E}_{x, y \sim \mathcal{D}_i} \exp\left(-1/2 \log\left(\frac{f^\star(x, y)}{\widehat{f}(x, y)}\right)\right)$$

$$\geq \frac{1}{2} \sum_{i=1}^n \mathbb{E}_{x \sim \mathcal{D}_i} \left\|\widehat{f}(x, \cdot) - f^\star(x, \cdot)\right\|_{\mathrm{TV}}^2.$$

Here the first identity uses the independence of the terms, which holds because $\widehat{f}$ is independent of the dataset $D'$. The second inequality is Lemma 22. This yields the theorem. $\qquad\square$

# E   Auxilliary Lemmas

**Lemma 23** (Elliptical Potential Lemma). *Consider a sequence of $d \times d$ positive semidefinite matrices $X_1, \ldots, X_T$ with $\max_t \mathrm{tr}(X_t) \leq 1$ and define $M_0 = \lambda I_{d \times d}, \ldots, M_t = M_{t-1} + X_t$. Then*

$$\sum_{t=1}^T \mathrm{tr}(X_t M_{t-1}^{-1}) \leq (1 + 1/\lambda) d \log(1 + T/d).$$

*Proof.* Observe that by concavity of the $\log \det(\cdot)$ function, we have

$$\log(\det(M_{t-1})) \le \log(\det(M_t)) + \operatorname{tr}(M_t^{-1}(M_{t-1} - M_t)).$$

Re-arranging and summing across all rounds $t$ yields

$$\sum_{t=1}^{T} \operatorname{tr}(X_t M_t^{-1}) \le \sum_{t=1}^{T} \log(\det(M_t)) - \log(\det(M_{t-1})) = \log(\det(M_T)) - d\lambda.$$

We will drop the negative term. By the spectral version of the AM-GM inequality and linearity of trace, we upper bound the last term:

$$\det(M_T)^{1/d} \le \operatorname{tr}(M_T)/d \le 1 + {}^{T}/d.$$

Now, we must convert from $M_t^{-1}$ to $M_{t-1}^{-1}$ on the left hand side. Fix a round $t$ and let us write $X_t = VV^\top$, which is always possible as $X_t$ is positive semidefinite. Then by the Woodbury identity

$$\begin{aligned}
\operatorname{tr}(X_t M_t^{-1}) &= \operatorname{tr}\left(V^\top (M_{t-1} + VV^\top)^{-1} V\right) \\
&= \operatorname{tr}(V^\top M_{t-1}^{-1} V) - \operatorname{tr}(V^\top M_{t-1}^{-1} V (I + V^\top M_{t-1}^{-1} V)^{-1} V^\top M_{t-1}^{-1} V).
\end{aligned}$$

All matrices are simultaneously diagonalizable, so we may pass to a common eigendecomposition. In particular, with the eigendecomposition $V^\top M_{t-1}^{-1} V = \sum_{i=1}^{d} \lambda_i u_i u_i^\top$, we obtain

$$\operatorname{tr}(X_t M_t^{-1}) = \sum_{i=1}^{d} \lambda_i - \frac{\lambda_i^2}{1 + \lambda_i} = \sum_{i=1}^{d} \frac{\lambda_i}{1 + \lambda_i} \ge \frac{1}{1 + {}^1/\lambda} \sum_{i=1}^{d} \lambda_i = \frac{1}{1 + {}^1/\lambda} \operatorname{tr}(X_t M_{t-1}^{-1}).$$

The inequality follows from the fact that $\lambda_i \le \left\| V^\top M_{t-1}^{-1} V \right\|_2 \le {}^1/\lambda$ due to our initial conditions on $M_0$ and the normalization for $X_t$. $\qquad\square$

**Corollary 24.** *Consider the setup of [Lemma 23](#) and further assume that for each $t$, we have $\operatorname{tr}(X_t M_{t-1}^{-1}) \ge \beta > 0$. Then $T \le 2(1 + {}^1/\lambda)d \log(1 + 2(1 + {}^1/\lambda)/\beta)/\beta$.*

*Proof.* The stated assumption and [Lemma 23](#) implies that $T \le (1 + {}^1/\lambda)d \log(1 + {}^T/d)/\beta$. We claim that if $T \le 2(1 + {}^1/\lambda)d \log(1 + 2(1 + {}^1/\lambda)/\beta)/\beta$ then a weakening of this bound is

$$\begin{aligned}
T &\le \frac{(1 + {}^1/\lambda)d}{\beta} \log(1 + {}^T/d)/\beta \le \frac{(1 + {}^1/\lambda)d}{\beta} \log\left(1 + \frac{2(1 + {}^1/\lambda)\log(1 + 2(1 + {}^1/\lambda)/\beta)}{\beta}\right) \\
&\le \frac{(1 + {}^1/\lambda)d}{\beta} \log\left(1 + \left(\frac{2(1 + {}^1/\lambda)}{\beta}\right)^2\right) \le \frac{2(1 + {}^1/\lambda)d}{\beta} \log\left(1 + \frac{2(1 + {}^1/\lambda)}{\beta}\right).
\end{aligned}$$

Therefore, we have established an upper bound on $T$. $\qquad\square$

## Footnotes

[9]Note that this is equivalent to the notion of backward kinematic inseparability (Misra et al., 2020).