[Reviews · NeurIPS 2020]

Review 1

Summary and Contributions: The paper proposes an algorithm for learning low rank MDPs with provable guarantees.

Strengths: The paper seems to be theoretical sound, well structured and highly coherent. The paper is highly relevant to the community, novel as far as I know.

Weaknesses: The significance of this paper is not entirely clear to me, while I understand why the authors chose not to provide simulations as the paper has theoretical merits, I can't help by wonder about its applicability. It's also interesting to think if and how the ideas and algorithms in the paper can combine with more modern RL algorithms.

Correctness: Yes.

Clarity: The paper is very well written - the authors seemed to put great effort in making the paper readable. Following the rebuttal - I agree with R2 that using h and H notation for stages is misleading as these letter are commonly used for episode count, and urge the authors to explain more clearly that the samples are taken from trajectories in their pseudo-code.

Relation to Prior Work: Yes, the authors related to several very recent works and provided detailed comparison.

Reproducibility: Yes

Additional Feedback:


Review 2

Summary and Contributions: The paper proposes a method for learning representations and utilizing that to promote exploratory behaviour in the finite-horizon low-rank MDP setting. The introduction highlights that despite the finite-horizon assumption low-rank MDPs can model MDPs that are classified as belonging to other classes -- like latent-variable MDPs and block MDPs. It shows that low-rank MDPs can effectively capture these variants, while being more efficient in terms of factorization for scalability. Following this, the proposed algorithm introduces two things: (1) a representation learning approach motivated by the factoring of low-rank MDPs as discussed in the paper, and (2) a reward-free exploration method to promote efficient model-learning in this finite-horizon scenario.

Strengths: The work is clearly motivated and the presentation of the framework, and its relationship to other works in literature that pursue similar representation learning goals is presented in a manner that is easy to follow. Section 4 is particularly well presented, and helps compare and contrast with respect to other work, and effectively shows the expressiveness of low-rank MDPs. While exploiting the low-rank structure of finite-horizon MDPs has gained the research community's attention lately, the work helps contextualize and relate these pieces, to propose a unifying representation learning framework. I think that is the main novelty of the work, and it would be useful to the active research community in the area.

Weaknesses: While the first half of the paper (till Section 5) is very clearly presented and easy to follow, Section 5 falls quite short on clarity. This is quite concerning as this is the section that contains the main proposal -- the algorithm, FLAMBE. I was unable to review much of the math, and analysis of the algorithm presented as it was fundamentally hard to understand the proposed algorithm. The notation in the pseudocode, and the text corresponding to it is surprisingly unclear. The objective for the Elliptical Planner, the sequence of steps in Algorithm 1, the enforcing of a uniform policy at every stage, are some factors of confusion.

Correctness: The paper mainly presents theoretical results on two fronts: (1) showing the effective expressive power offered by low-rank MDPs, and (2) the efficiency of the proposed algorithm, FLAMBE, for low-rank MDPs. While the material presented to ascertain the case for (1) were clear and easy to follow, and hence easy to judge correctness of, the presentation of (2) was rather hard to understand and consequently hard to evaluate the correctness of. There are no empirical components to the content in the paper.

Clarity: As mentioned earlier, the first part of the paper was very clear. The presentation of the algorithm, details corresponding to it, and consequently the theory that follows were unclear, and need to be improved. As presented the main proposal of the paper is quite inaccessible, and it may be very beneficial to clarify that in the following submission or rebuttal.

Relation to Prior Work: Yes, the work is contextualized quite well. But as representation learning for RL is quite a vast topic, there is explicit work in the literature that focuses on this in the infinite-horizon setting. In this regard, I think the authors cite Pathak et. al. 2017 and Tang et. al. 2017 which is great; but representations here are primarily motivated in their utility for exploration. So a fundamental work in this regard is [1]. Additionally many other works exist which focus on learning effective representation in the RL setting for infinite-horizon case like [2,3,4]. I think some, if not all, can be included to create a more comprehensive related works section. [1] Bellemare, Marc, Sriram Srinivasan, Georg Ostrovski, Tom Schaul, David Saxton, and Remi Munos. "Unifying count-based exploration and intrinsic motivation." In Advances in neural information processing systems, pp. 1471-1479. 2016. [2] Wu, Yifan, George Tucker, and Ofir Nachum. "The Laplacian in RL: Learning representations with efficient approximations." arXiv preprint arXiv:1810.04586 (2018). [3] Liu, Vincent, Raksha Kumaraswamy, Lei Le, and Martha White. "The utility of sparse representations for control in reinforcement learning." In Proceedings of the AAAI Conference on Artificial Intelligence, vol. 33, pp. 4384-4391. 2019. [4] Barreto, André, Will Dabney, Rémi Munos, Jonathan J. Hunt, Tom Schaul, Hado P. van Hasselt, and David Silver. "Successor features for transfer in reinforcement learning." In Advances in neural information processing systems, pp. 4055-4065. 2017.

Reproducibility: No

Additional Feedback: Other than the comments and suggestions made above, here are the key points of confusion with regards to the presentation of FLAMBE, which can serve as a basis for improving the presentation of it: - loop over episodes missing: is the algorithm just run for an episode? - sequence of steps in Algorithm 1: the MLE problem provides a \phi_h and \mu_h. This is necessary for the planning step to propose a policy \pi_h. But what exactly is the policy followed for data collection in horizon step h before MLE? - Is access to a generative model assumed? If not, how exactly is the data obtained for utilizing in the MLE estimation step? - Where is the SAMP oracle used? - what is the point of enforcing uniformity of the "exploratory policy"? - why uniformity enforcement twice? (beginning and end of loop) - Although T is learned for each step of the horizon, the planner only needs access to the latest T_h for planning \pi_h I think. But this is unclear. - I do not see how Algorithm 2 corresponds to Jin et. al.'s optimistic-LSVI algorithm. The pseudocode says "see text for details", but there are no details in the text. What is \pi_0? Optimistic-LSVI performs regression to learn w, but here a policy \pi is learned? What is the parameter \beta? Also, \argmax used, but objective evaluation necessary for checking the condition. - Considering the algorithm is run only once, what is the utility of the "exploratory" policy? - The averaging used for updating \Sigma_t -- just a cumulative sum, no averaging? Given these confusions I was unable to evaluate the theory that follows. While I understand addressing each point of confusion in the rebuttal may be hard, if the proposal and the exact assumptions made, can be summarized succinctly and clearly in the rebuttal, that'd be extremely useful. In any case, I think the work is interesting and I hope to see it refined and published in the near future. Minor comments: - Line 23: "downstream tasks" --- multiple tasks? I think downstream tasks is a consequent of having a main task (as used in transfer learning/auxiliary tasks literature). Representation learning on the other hand is a task, the output of which we aim to use improve efficiency of learning for many tasks, but they are not necessarily "downstream". - Line 311: roll->role --- Post-rebuttal --- Thank you for the many clarifications, and I apologize for my confusions! I see that I poorly understood the proposed Elliptical Planner, and I now appreciate the novelty of it for learning models in a reward-free framework. But I'd like to re-emphasize that I think the pseudocode can be improved to be clearer -- particularly for sampling "n" trajectories at a given horizon, I'm unsure of the policy followed from h:H-1. I hope that can be cleared up in the code or via description to make the paper more accessible. With this, I'm increasing my score to an accept with a reduced confidence. PS: In the context of VAML work mentioned by R4, a recent work which utilizes similar ideas for learning effective models in the with-reward setting may be of interest here as well: Ayoub, Alex, et al. "Model-Based Reinforcement Learning with Value-Targeted Regression." arXiv preprint arXiv:2006.01107 (2020).


Review 3

Summary and Contributions: The paper provides a theoretical study of representation learning and exploration for low rank MDPs. The paper also develops FLAMBE, a computationally and statistically efficient model based algorithm for representation learning in low rank MDPs.

Strengths: 1. The paper proves that the expressiveness of low rank MDPs is better than block MDPs. 2. The paper develops an algorithm, FLAMBE, to learn the representation for low rank MDPs.

Weaknesses: 1. The paper does not follow the conference reference style. Please update the references following the guidelines. 2. The complexity of the content requires more pages allowed by the conference. Therefore, some important discussions and proofs are served in the Appendix. This leads to some inconvenience in reading. 3. It's not clear if Assumption 1 is also necessary for block MDPs. Please provide a clear statement and discussion in the paper. 4. As the work is theoretical, it is unclear how it performs and compares with other work in representation learning in terms of computational efficiency and result accuracy/effectiness. A discussion of one or two paragrahs is necessary. 5. The paper shows that low rank MDPs are significantly more expressive than the block MDP model. It is interesting to include a discussion on the boundary of low rank MDPs regarding the expressiveness.

Correctness: Yes

Clarity: Yes

Relation to Prior Work: It seems so.

Reproducibility: Yes

Additional Feedback:


Review 4

Summary and Contributions: This paper offers a computationally efficient algorithm, FLAMBE, for learning an approximate transition model within a finite-horizon MDP. The algorithm operates in the absence of any particular reward function (the reward-free setting) and accommodates possibly non-stationary MDP transition dynamics. The authors center their algorithm and analysis around a particular (but not novel, L105-107) structural assumption, designated a low rank MDP, where individual transition probabilities decompose into the inner product of two embeddings derived from the current state-action pair and next state, respectively. In adopting this structure, the authors aim to design a provably-efficient algorithm for learning the underlying features that give way to an epsilon-approximate transition model (in total variation distance). The use of the reward-free exploration setting is justified by showing accurate linear Q-function approximation using the learned embedding functions as features (Lemma 1). The authors connect their low rank assumption to latent variable graphical models, illustrating the theoretical implications of this (Propositions 1 & 2). Under assumptions of realizability and reachability (in the latent feature space), the authors offer an explicit sample complexity guarantee for FLAMBE (Theorems 2 & 3). The authors' response hasn't changed my view of the paper.

Strengths: This paper develops a provably-efficient algorithm for the challenging task of recovering a good approximation to the MDP transition function, improving upon the computational inefficiencies or strong assumptions of prior work (L267-271). Central to this problem is the challenge of exploration for achieving good state-action space coverage; the elliptical planning algorithm employed by FLAMBE (Algorithm 2) seems novel and is a contribution on its own. The authors ground their structural assumption to latent-variable graphical models, creating an opportunity for recent progress at the intersection of neural networks and probabilistic inference to fuel practical, scalable instantiations of FLAMBE.

Weaknesses: For completeness, it would be nice to contrast the contributions here against classic work on state abstraction in RL, specifically bisimulation [4,5,6,7]. The paper could use more motivation/clarity on the goal of system identification (as given by Eqn. 1) and the reward-free learning setting. I believe these two things are actually coupled; there are papers [1,2,3] which note that mastering every last detail of the transition function may be an exaggeration of the actual challenge faced by an agent with a specific reward function. In other words, depending on the downstream planning algorithm and/or reward function(s) presented after learning the transition model, it maybe be the case that FLAMBE incurs greater sample complexity than a method that waits to leverage the full structure of the problem. Conversely, there are likely conditions on the subsequent reward functions (e.g. sparsity) that warrant the system identification goal of FLAMBE. As a concrete example, note that in a MDP which models some real-world process, the objective of Eqn. 1 amounts to learning an approximate physics engine, something that might be computationally intractable for even the most generous settings of epsilon. Some discussion of this kind around the objective of full system identification and the potential (sample complexity) cost of reward-free learning might highlight subsequent areas for theoretical analysis and/or identify practical applications that may be best suited to FLAMBE. The results concerning Bellman/Witness rank, while nominally interesting, seem out of place, bordering on superfluous. Perhaps these bits of the main paper could be relegated to the appendix and the space reallocated for more discussion of related work, open problems/future work, etc.? [1] Farahmand, Amir-massoud, Andre Barreto, and Daniel Nikovski. "Value-aware loss function for model-based reinforcement learning." In Artificial Intelligence and Statistics, pp. 1486-1494. 2017. [2] Farahmand, Amir-massoud. "Iterative value-aware model learning." In Advances in Neural Information Processing Systems, pp. 9072-9083. 2018. [3] Nair, Suraj, Silvio Savarese, and Chelsea Finn. "Goal-Aware Prediction: Learning to Model What Matters." arXiv preprint arXiv:2007.07170 (2020). [4] Li, Lihong, Thomas J. Walsh, and Michael L. Littman. "Towards a Unified Theory of State Abstraction for MDPs." In ISAIM. 2006 [5] Dean, Thomas, and Robert Givan. "Model minimization in Markov decision processes." In AAAI/IAAI, pp. 106-111. 1997. [6] Givan, Robert, Thomas Dean, and Matthew Greig. "Equivalence notions and model minimization in Markov decision processes." Artificial Intelligence 147, no. 1-2 (2003): 163-223. [7] Ferns, Norm, Prakash Panangaden, and Doina Precup. "Metrics for Finite Markov Decision Processes." In UAI, vol. 4, pp. 162-169. 2004.

Correctness: I think the results are correct.

Clarity: The paper is well written

Relation to Prior Work: Yes.

Reproducibility: Yes

Additional Feedback:

[Author Response · NeurIPS 2020]

Thanks for the reviews! Response to individual reviewers are below.

**R1.** You are right that the contribution is theoretical, but our formulation of the representation learning problem and the computational oracles assumed are heavily motivated by practice. The FLAMBE algorithm is not difficult to implement with modern deep learning libraries. In addition, modern RL algorithms can be employed, in the planning step (Eq. (2) in Algorithm 2, see e.g. the recent preprint `https://arxiv.org/abs/2007.08459`), as well as in the maximum likelihood step (e.g., using VAEs as in recent model-based RL works such as `https://arxiv.org/abs/2005.05960`). To summarize, our theory directly motivates new strategies for representation learning and exploration that we believe will be empirically effective, and we are excited about experimenting with these approaches in future work.

**R2.** We believe the algorithm is actually quite precisely written and intuitive. Perhaps the simpler algorithm (and simpler analysis) for the simplex features setting provided in the appendix is useful for added intuition. The algorithm at a high-level can be seen as doing MLE to estimate features, and using an algorithm for low-rank MDPs with known features to compute a good policy. Inducting these two at each step $h = 0, \ldots, H$ with some careful attention to details gives the FLAMBE algorithm. For specifics:

- We believe you may have missed the role of the exploratory policy $\rho_h$ which is computed in iteration $h - 1$ of the algorithm (via a call to Algorithm 2) and is used to collect the data for the MLE step in iteration $h$. Concretely, our model-based planning step (call to Alg 2) returns an exploratory policy $\rho_h^{\text{pre}}$, which is executed for $h - 1$ steps, followed by *two* uniform actions at steps $h$ and $h + 1$ to collect the samples for MLE estimation at level $h + 1$. We need two random actions as planning stays one step behind model learning, as briefly discussed in lines 291–295. We *do not* assume access to a generative model; we are in the online exploration setting and collect data by the learned exploratory policies $\rho_h$. The use of such exploratory policies to cover states well is common to many provable methods (e.g., Du et al, 2019b, Misra et al., 2019).
- Loop over episodes missing – The only interaction with the environment happens in the line "Collect n triples ..." in Algorithm 1. To collect one triple, we execute a full episode, so the total number of episodes is $nH$.
- Objective for elliptical potential – This quadratic objective originated in the linear bandits literature and is quite standard in linear RL problems with known features, e.g., it appears in the analysis of Jin et al.
- The sampling oracle is used in Algorithm 2, to optimize Eq (2) in the learned model. This can be done with dynamic programming-type algorithms, such as Optimistic LSVI, which is quite straightforward and described in Appendix C, Lemma 5. We *do not* say that Algorithm 2 corresponds to optimistic LSVI.

Thanks for the references. We agree the count-based exploration work is relevant. Note that the other works mentioned do not consider representation learning in the context of exploration, which is our focus, and so they are less relevant. We will add more discussion to address some of the confusions above,

**R3.** Assumption 1 in block MDPs – You are right, the realizability assumptions in our work and block MDP results are not equivalent, but there is some subtlety here. We consider "model-based realizability" while a weaker notion may be simply that $\phi^\star \in \Phi$, with no assumption on $\mu^\star$. The realizability assumption for block MDP results is between these two: Du et al.'s and Misra et al.'s assumption is equivalent to realizability of $\phi^\star$ and *the support* of $\mu^\star$ (since the next state is also decodable). We do not see a natural analog to this intermediate assumption for the general low rank setting, but we agree that considering the weaker assumption (only $\phi^\star \in \Phi$) is a nice question for future work. Note also that when we discuss expressiveness, we are focusing only on the dynamics assumptions, and not on realizability requirements. We can expand on this in the final version.

We can add some discussions around practical issues, comparison to other works, and on limitations of low rank MDPs.

**R5.** Thanks for the pointers to bisimulation! We can discuss more in the final version. Briefly, our learned representations are related to a state abstraction notion, called Kinematic Inseparability (from Misra et al. 2019), which is finer than bisimulation, but remains meaningful in the reward-free setting. Note that in the absence of rewards, every MDP admits a trivial bisimulation that aggregates all states together. Indeed, it is not possible to learn a bisimulation with polynomial sample complexity while exploring in a sparse reward problem, as formally proved in Modi et al. 2020 (Proposition B.1). Thus our abstraction is less coarse, but learnable in the exploration setting.

Regarding reward-free exploration: You are right that system identification may be overkill for easy RL problems (e.g., dense rewards, local exploration suffices, etc.). However, we are interested in *provable* sample efficiency, which means we must consider *hard* RL problems. In this case, the distinction between reward-free and reward-sensitive learning is less significant, since even in the reward-sensitive setting the agent must explore the entire environment so it can certify that there is no "hidden rewards" anywhere.

You are also hinting at a separation between model-based and value/policy-based algorithms. This seems plausible, and we agree that developing provable model-free algorithms for low rank MDPs is an exciting direction for future work.

[Meta-Review · NeurIPS 2020]

The reviewers all agree that this work is solid, with important novel insights and clear writing. For a final paper, do carefully consider reviewer suggestions to improve the paper.